THE
EMBO
JOURNAL

# The coordinated action of UFMylation and the RQC pathways clears arrested polypeptides at the ER

Milica Mihailovic [iD] [1,2,7], Aleksandra S Anisimova [iD] [1,2,7], Bu Erte [iD] [3], Ni Zhan[3], Ioanna Styliara[1,4], Yasin Dagdas [iD] [3,5 ✉] & Gülsün Elif Karagöz [iD] [1,6 ✉]

## Abstract

Clearance of arrested nascent polypeptides resulting from ribosomal stalling is essential for proteostasis. Stalled endoplasmic reticulum (ER)-bound ribosomes are marked by ubiquitin-fold modifier 1 (UFM1) on the large ribosomal subunit protein RPL26, but the precise role of this modification in ribosome-associated quality control (RQC) remains poorly understood. Here, we define the interplay between the UFMylation machinery and the RQC in clearing arrested polypeptides upon ribosome stalling at the ER. Proteomic analysis shows that RQC factors associate with UFMylated ribosomes. Functional assays demonstrate that ribosome rescue factors ZNF598 and ASC-1 recognize and split stalled ribosomes at the ER, a prerequisite for RPL26 UFMylation. The UFM1 E3 ligase complex then binds and UFMylates the post-split 60S-peptidyl-tRNA complex, facilitating access of RQC factors. Depletion of the NEMF/LTN1 complex leads to accumulation of UFMylated ribosomes, whereas impaired UFMylation weakens NEMF/LTN1 binding to ER-stalled ribosomes, supporting a physical link between these pathways. These findings demonstrate that RQC cooperates with the UFMylation machinery to overcome the topological constraints of clearing the arrested polypeptides at the ER.

Keywords UFMylation; Ribosome-associated Quality Control; Translation; Endoplasmic Reticulum; Ribosome Stalling
Subject Categories Post-translational Modifications & Proteolysis; Translation & Protein Quality

## Introduction

To maintain cellular protein homeostasis, protein translation is constantly monitored by quality control factors. Distinct secondary mRNA structures, incompletely processed mRNAs, rare codons, or translation into poly(A) tail can lead to prolonged translational pausing and ribosomal stalling (Shoemaker and Green, 2012; Choe et al, 2016; Doma and Parker, 2006; Letzring et al, 2013; Juszkiewicz and Hegde, 2017). Like a traffic jam, stalled ribosomes lead to ribosomal collisions and impaired translation (Chandrasekaran et al, 2019; Tesina et al, 2020). Also, arrested nascent polypeptides are prone to aggregation, which perturbs homeostasis (Choe et al, 2016). The highly conserved ribosome rescue and ribosome-associated quality control (RQC) pathways recognize and recycle stalled ribosomes and degrade incomplete nascent chains to maintain proteostasis (Filbeck et al, 2022).

The mechanisms involved in clearing stalled ribosomes in the cytosol have been studied in depth (reviewed (Joazeiro, 2017, 2019; Filbeck et al, 2022; Inada and Beckmann, 2024)). Collided ribosomes are recognized by the E3 ligase ZNF598, which ubiquitinates 40S ribosomal proteins and halts translation (Juszkiewicz and Hegde, 2017; Sundaramoorthy et al, 2017; Juszkiewicz et al, 2018). This is followed by the binding of the ribosome splitting factors ASC-1 complex/Pelota that recognize ubiquitinated stalled ribosomes (Juszkiewicz et al, 2020; Hashimoto et al, 2020) and split the leading ribosome to release the 60S-peptidyl-tRNA complex. After splitting, the 60S-peptidyl-tRNA complex is recognized by the RQC machinery, NEMF/LTN1 complex. NEMF adds template-independent C-terminal alanine and threonine extensions (CAT-tails) to the stalled nascent chain, which facilitates the exposure of the nascent chains out of the ribosome tunnel for ubiquitination by the E3 ligase LTN1 (Shen et al, 2015; Bengtson and Joazeiro, 2010; Kostova et al, 2017; Chu et al, 2009; Lyumkis et al, 2013). Subsequently, the nascent chain is released from the ribosome after tRNA cleavage by ANKZF1, extracted from the 60S ribosomal subunit by AAA ATPase VCP, and finally targeted for proteasomal degradation (Kuroha et al, 2018; Brandman et al, 2012; Meyer et al, 2012). The released 60S subunit is recycled and ready for translation initiation.

Translation into the endoplasmic reticulum (ER) membrane or the ER lumen poses topological and steric challenges for the RQC machinery to access the nascent chains on ER-bound ribosomes upon stalling. These nascent chains also obstruct the translocon, hindering the synthesis and maturation of other proteins at the ER, which causes an additional burden on proteostasis (Izawa et al, 2012; Phillips and Miller, 2020). Recent studies showed that

[1]Max Perutz Laboratories Vienna, Vienna BioCenter, Vienna, Austria. [2]Vienna BioCenter PhD Program, Doctoral School of the University of Vienna and Medical University of Vienna, Vienna, Austria. [3]Gregor Mendel Institute (GMI), Vienna BioCenter, Vienna, Austria. [4]University of Vienna, Vienna, Austria. [5]Heidelberg University, Centre for Organismal Studies (COS), Heidelberg, Germany. [6]Medical University of Vienna, Vienna, Austria. [7]These authors contributed equally: Milica Mihailovic, Aleksandra S Anisimova.
✉E-mail: yasin.dagdas@cos.uni-heidelberg.de; guelsuen.karagoez@meduniwien.ac.at

ribosomes stalled at the ER are marked by UFM1 (ubiquitin-fold modifier 1) at the large ribosomal subunit protein RPL26 (Wang et al, 2020; Walczak et al, 2019), and impairing UFMylation results in the accumulation of arrested polypeptides at the ER (Scavone et al, 2023). Similar to ubiquitination, UFMylation proceeds through a cascade of enzymatic reactions catalyzed by an E1-activating enzyme (UBA5), an E2-conjugating enzyme (UFC1), and an E3-ligating enzyme complex (UFM1 E3 ligase complex) (Kumar et al, 2021; Komatsu et al, 2004; Gavin et al, 2014; Stephani et al, 2020; Tatsumi et al, 2010). The UFM1 E3 ligase complex consists of three components: UFL1, DDRGK1, and CDK5RAP3 (C53), and is tethered to the ER by the ER-membrane protein DDRGK1. UFMylation is essential for ER homeostasis, and it is implicated in ER-phagy and RQC (Wang et al, 2020; Scavone et al, 2023; Stephani et al, 2020). Yet, due to a lack of biochemical and structural work, our understanding of the clearance of the ER-bound ribosomes that co-translationally translocate nascent chains into the ER remained incomplete (Karagöz et al, 2019; Lakshmi-narayan et al, 2020). More specifically, it is unclear whether UFMylation and RQC directly interact or act independently.

Accumulating evidence supports two models, not necessarily mutually exclusive, that explain the role of UFMylation in ER-localized translation. The first model suggests that ribosomal stalling at the ER induces RPL26 UFMylation, and UFMylation is involved in the clearance of arrested polypeptides resulting from ribosomal stalling (Wang et al, 2020; Scavone et al, 2023). The second, more recent model proposes that UFMylation releases the 60S ribosomal subunit from the translocon following the canonical translation termination at the ER (DaRosa et al, 2024; Makhlouf et al, 2024). The second model emerged from recent structural work showing that the UFM1 E3 ligase complex binds to an empty 60S subunit that sterically clashes with the translocon and a tRNA at the P-site. However, this novel binding mode cannot explain how the depletion of the UFMylation machinery leads to the accumulation of arrested polypeptides in the cell.

To reveal how UFMylation machinery clears arrested peptides at the ER, we dissected the interplay between the RQC and UFMylation machinery in this process. We show that the UFM1 E3 ligase complex loads onto the 60S-peptidyl-tRNA complex after ribosomal splitting by ribosome rescue factors. Through close collaboration with the RQC factors NEMF/LTN1, the UFM1 E3 ligase complex facilitates the clearance of stalled peptides by enabling better access of RQC factors to the nascent chain. Altogether, our findings underline the crucial role of the UFMylation-RQC crosstalk for clearing arrested polypeptides at the ER.

## Results

### RQC factors associate with and act on UFMylated ribosomes

To discover the pathways involved in the clearance of UFMylated ribosomes, we isolated them from mammalian cells for mass spectrometry (MS) analysis. To efficiently pull down UFMylated ribosomes, we generated an HCT116 cell line with endogenously tagged FLAG-UFM1 using the CRISPR-Cas9 gene editing approach. We enriched ribosomes by pelleting them using sucrose

cushion sedimentation under control conditions or after inducing ribosomal stalling with anisomycin (ANS), which impairs ribosomal peptidyl transferase activity (Hansen et al, 2003). UFMylated ribosomes were subsequently pulled down by FLAG immunoprecipitation and subjected to MS (Fig. 1A). The MS analyses showed enrichment of UFM1 E3 ligase complex components (UFL1, CDKRAP53, DDRGK1) in FLAG-immunoprecipitated samples from FLAG-UFM1 cells treated with anisomycin, compared to the IgG control, and to untreated cells as a control (Fig. 1B,C; complete list in Dataset EV1). In addition to the components of the UFM1 E3 ligase complex, we found that the RQC component NEMF specifically associated with UFMylated ribosomes in a stalling-dependent manner (Fig. 1B,C). FLAG-UFM1 IPs followed by western blot analyses showed efficient enrichment of UFMylated RPL26 in the pulldowns relative to lysates and to non-modified RPL26, which further increased upon ribosomal stalling (Fig. 1D,E). Under those conditions, we observed a stalling-dependent association of UFM1 E3 ligase components, as well as NEMF and LTN1 in FLAG-UFM1 pulldowns, confirming our MS results (Figs. 1D,E and EV1A). IP of Sec61β in control conditions and upon ANS treatment showed stalling-dependent increase in association of UFMylated RPL26 with the translocon, supporting the notion that ER-bound ribosomes are UFMylated (Fig. EV1B).

NEMF/LTN1 associates with the peptidyl-tRNA-60S after splitting of stalled ribosomes (Shao et al, 2015; Shao and Hegde, 2014). To assess whether NEMF and UFL1 co-migrate at similar ribosomal fractions, we performed polysome profiling. Supporting structural and biochemical data, NEMF/LTN1 migrated primarily with the 60S ribosomal subunit in polysome profiles. Instead, UFL1 showed a broad distribution across 40S, 60S, and 80S fractions in two cell lines (HCT116 and HEK293T; Figs. 1F,G and EV1C). Notably, unlike UFL1, the majority of UFM1-conjugated RPL26 co-migrated with the 60S subunit, with a small fraction co-migrating with the 80S, as shown previously (Wang et al, 2020; DaRosa et al, 2024) (Figs. 1F,G and EV1C). This is consistent with our FLAG-UFM1 pulldowns, in which small ribosomal proteins were not enriched (Figs. 1E and EV1A). Additionally, we detected ZNF598-dependent ubiquitination of 40S ribosome protein RPS10 on disomes by polysome profiling in HCT116 cells treated with ANS. At the same time, UFMylation was mainly enriched in the 60S fractions, suggesting that early ZNF598 ubiquitination and UFMylation target distinct substrates (Fig. EV1D).

The association of RQC components with the UFMylated ribosomes supports their role in clearing ribosomes stalled at the ER. Together, these findings suggest a crosstalk between the RQC and the UFMylation machinery in resolving stalled ribosomes at the ER.

### Ribosome rescue factors recognize and mark ER-stalled ribosomes

To further assess the functional role of ribosome rescue factors in recognizing and recycling the ribosomes stalled at the ER, we monitored the translational readthrough of well-characterized stalling cytosolic or ER-targeted reporters (Wang et al, 2020) (Fig. 2A) in a fluorescence-activated cell sorting (FACS) assay (Fig. 2B,C). The ERK20 reporter consists of an N-terminal signal sequence, an N-glycosylation site, followed by EGFP and a poly-

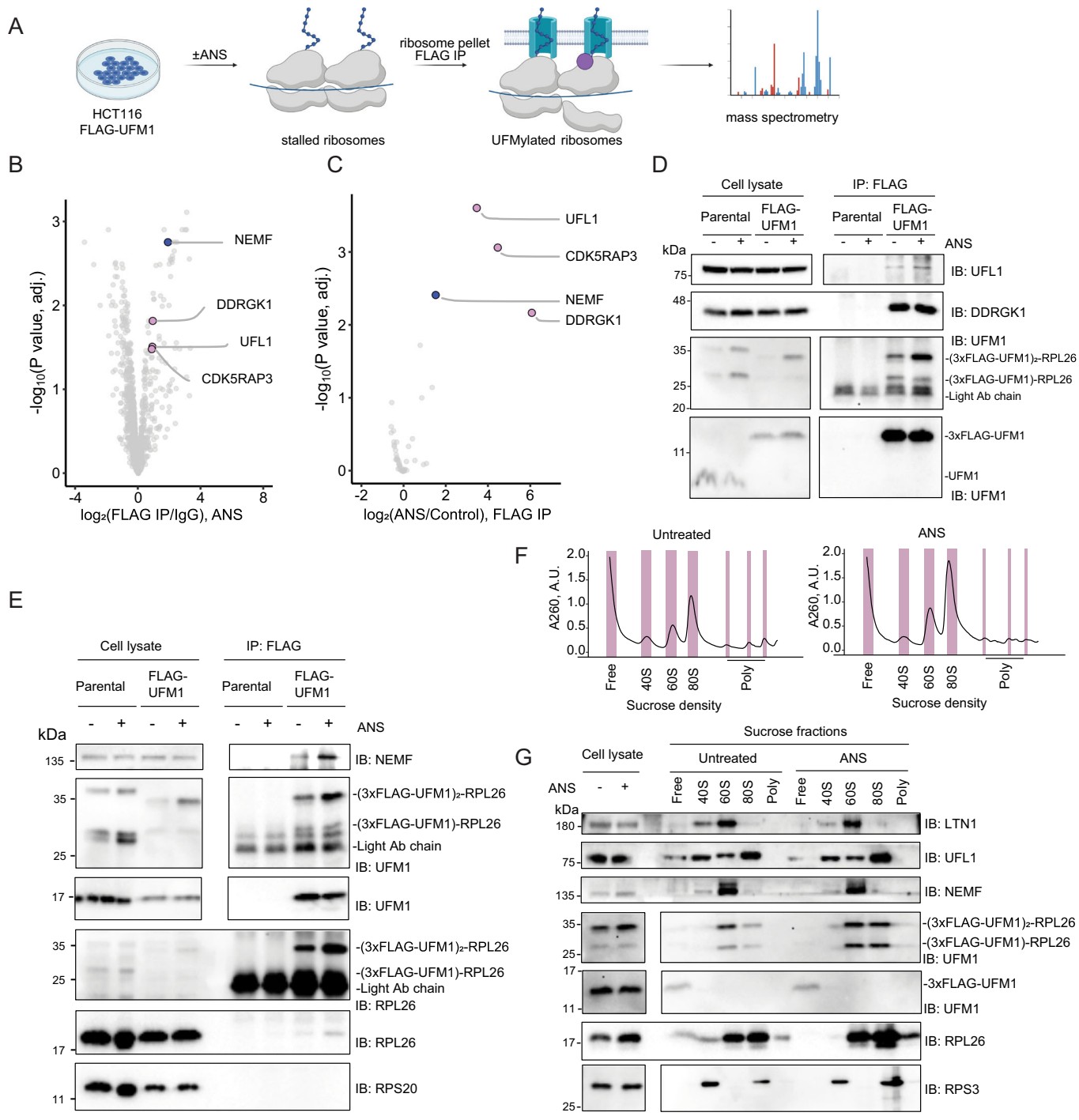

**Figure 1. RQC factors directly associate with UFMylated ribosomes.**

(A) Scheme of mass spectrometry analysis of UFMylated ribosomes. (B) Volcano plot of proteins co-enriched with UFMylated ribosomes in FLAG IP over control IgG IP upon 1 h 200 nM ANS treatment in HCT116 FLAG-UFM1 cells, identified by mass spectrometry experiment from (A), $n = 3$. (C) Volcano plot of enrichment of FLAG-eluates from ANS-treated HCT116 FLAG-UFM1 cells over FLAG-eluates from untreated control HCT116 FLAG-UFM1 cells, identified by mass spectrometry experiment from (A), $n = 3$. Proteins enriched 1.5 times in FLAG IP over IgG IP in ANS condition (P value adj. <0.05) are shown, calculated with limma-moderated Benjamini–Hochberg-corrected two-sided $t$-test. (D, E) Immunoblot analysis of FLAG IP eluates upon 200 nM 1 h ANS treatment or from untreated HCT116 parental or HCT116 FLAG-UFM1 cells. (F) Polysome profiling of HCT116 FLAG-UFM1 cell lysates upon 200 nM 1 h ANS treatment or from untreated control condition. (G) Immunoblot analysis of selected sucrose fractions from (F). Source data are available online for this figure.

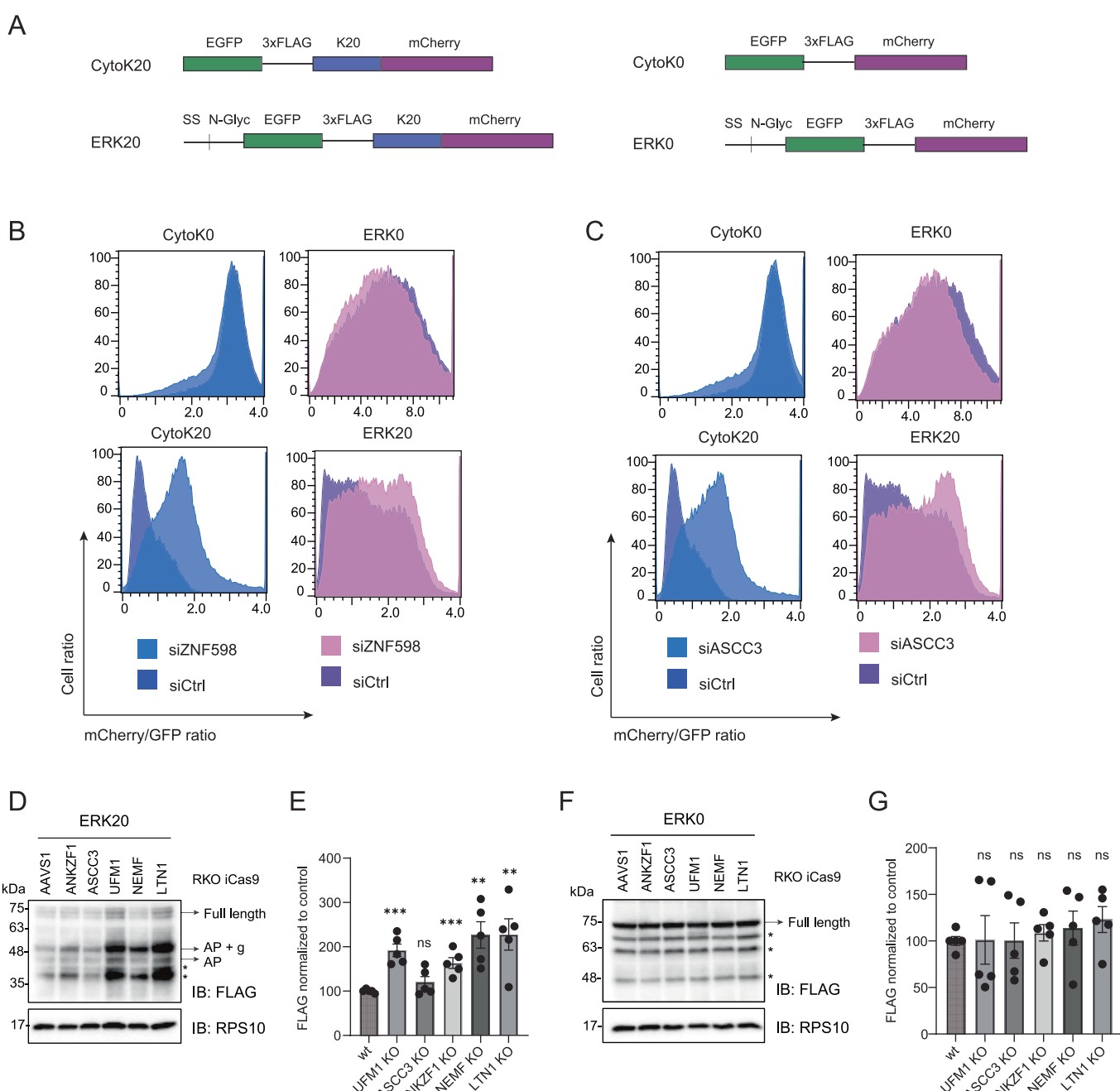

**Figure 2. RQC and UFMylation machinery are needed to clear stalled peptides at the ER.**

(A) Schematic representation of control (K0) and stalling (K20) cytosolic and ER reporters. (B, C) Readthrough of reporters from (A) shown by mCherry/GFP ratio measured by FACS after 24 h expression in HCT116 cells upon siRNA-mediated knockdown of ZNF598 (B) or ASCC3 (C) compared to non-targeting control siRNA. (D, F) Reporter accumulation shown by FLAG immunoblot in RKO iCas9 KO cell lysates upon 48 h dox treatment to induce AAVS1 (non-targeting control), ANKZF1, ASCC3, UFM1, NEMF or LTN1 knockout, upon 24 h ERK20 (D) or ERK0 expression (F) normalized to RPS10 loading control. (E) Quantification of (D) with unpaired two-sided Student $t$-test, $n = 5$. Error bars represent standard error of the mean (SEM). ****$P \leq 0.0001$, ***$P \leq 0.001$, **$P \leq 0.01$, *$P \leq 0.05$. List of complete $P$ values available in Dataset EV2. (G) Quantification of (F) with unpaired two-sided Student $t$-test, $n = 5$. Error bars represent SEM. AP arrested peptide, g glycosylation, * degradation products. Source data are available online for this figure.

lysine stretch that mimics translation into polyA tail, and a C-terminal mCherry. In contrast, the cytosolic stalling reporter CytoK20 lacks the ER-targeting signal sequence and glycosylation site, while having the same domain organization and sequence (Fig. 2A). We used siRNA to deplete ZNF598 and ASCC3, a component of the ASC-1 complex (Fig. EV2A) as the depletion of ASCC3 leads to the 80% destabilization of the whole complex (Juszkiewicz et al, 2020). Both the cytosolic (as shown earlier (Juszkiewicz et al, 2018)) and ER stalling reporters exhibited increased readthrough upon loss of ZNF598 and ASCC3 (Fig. 2B,C).

In contrast, control cytosolic CytoK0 and ER-targeted ERK0 reporters, which lack the stalling-inducing poly-lysine sequence, did not show a difference in readthrough upon depletion of the ribosome rescue factors (Fig. 2B,C). Importantly, the loss of Pelota, which acts specifically on ribosomes stalled at the 3'-end of the RNA (Izawa et al, 2012; Guydosh and Green, 2014), did not impact the readthrough of our internal ribosomal stalling reporters, validating our findings (Fig. EV2B). These results suggest that, as observed for cytosolic ribosomes, the ribosome rescue factors ZNF598 and the ASC-1 complex are involved in the recognition and splitting of ribosomes stalled at the ER, indicating that they can access and act on the ER-bound ribosomal pool.

## Both the RQC and the UFMylation machinery clear stalled polypeptides at the ER

After demonstrating that ribosome rescue components are involved in the clearance of ER-stalled ribosomes, we next investigated the interplay among ribosome rescue, RQC, and the UFMylation machinery in the clearance of the model ER stalling substrate, ERK20. To deplete those factors efficiently, we used RKO cell lines that stably express doxycycline-inducible Cas9 (iCas9) (de Almeida et al, 2021). Using lentiviral transduction, we introduced either specific guide RNAs (gRNAs) targeting the ribosome rescue and RQC components (ANKZF1, ASCC3, NEMF, LTN1) or the non-coding control locus AAVS1. In parallel, we used UFM1 iCas9 cells to impair UFMylation. Treatment of cells with doxycycline for 48 h resulted in efficient depletion of the ribosome rescue and RQC factors or UFM1 (96% ANKZF1, 71% ASCC3, 75% NEMF, 69% LTN1, and 93% UFM1, Fig. EV2C). Supporting earlier evidence (Wang et al, 2020; Scavone et al, 2023), the depletion of NEMF, LTN1, and UFM1 significantly stabilized the ER stalling reporter ERK20 (Fig. 2D,E). Moreover, while less efficiently, depletion of ANKZF1 also led to stabilization of ERK20 (Fig. 2D,E). Instead, depletion of ASCC3 did not impact the levels of the ERK20 reporter. This is expected since the ASC-1 complex mediates ribosome splitting, and its depletion results in less efficient stalling and increased read-through. Importantly, depletion of any of these genes did not cause accumulation of the control ER reporter ERK0 (Fig. 2F,G), validating the direct role of these factors on the clearance of arrested polypeptides generated by ribosome stalling at the ER.

Next, we tested the downstream degradation pathway of cytosolic and ER stalling reporters through proteasomal inhibition by MG132 or inhibition of lysosomal degradation by bafilomycin treatment. Both reporters were stabilized by the inhibition of the proteasome and lysosomal degradation machinery. Yet, the proteasomal inhibition showed a higher accumulation of the reporters, indicating its major contribution to their clearance (Fig. EV2D). Taken together, we show that the UFMylation and the RQC machinery are necessary for clearing arrested nascent peptides on stalled ribosomes at the ER.

## Recognition and splitting of the ER-stalled ribosomes precedes RPL26 UFMylation

Next, we investigated the crosstalk between the RQC and UFMylation. To reveal the order of events and the interdependence of these pathways, we first assessed whether the loss of ribosome

rescue or RQC components impacts UFMylation of the ER-stalled ribosomes. To this end, we used RKO iCas9 cell lines expressing gRNAs against those components (Fig. EV2C). Under steady-state conditions, depletion of ZNF598, ASCC3, or Pelota did not impact UFMylation levels (Figs. 3A, lanes 1–4 and 3B). However, upon ANS treatment, we noticed a significant decrease in UFMylation levels upon depletion of ZNF598, ASCC3, or Pelota compared to control (AAVS1) (Figs. 3C, lanes 1–4 and 3D). These results were corroborated in the HCT116 cell line using siRNA-mediated depletion (Fig. EV3A,B). These data indicate that ribosome splitting is required for UFMylation of ribosomes at RPL26 after ribosomal stalling. Recent work showed that RPL26-UFMylation also occurs after canonical translation termination at the ER, releasing free 60S from the translocon (DaRosa et al, 2024; Makhlouf et al, 2024). We anticipate that RPL26 UFMylation at steady-state conditions is governed by two main events: i. translation termination at the ER following ribosome splitting via translation termination factors, and ii. ribosomal stalling at the ER followed by ribosomal splitting via ASC-1 or Pelota, depending on the type of mRNA lesion. Therefore, the depletion of ASCC3 or ZNF598 has only a partial effect in our assays.

Notably, the depletion of RQC factors NEMF and LTN1, which facilitate ubiquitination and clearance of the nascent chain, increased RPL26 UFMylation levels under steady-state conditions (Figs. 3A, lanes 5, 6 and 3B), suggesting that RQC continuously surveils aberrant translation, supporting earlier work that revealed that LTN1 depletion impacts RPL26 UFMylation in neurons (Endo et al, 2023). We anticipate that depletion of NEMF and LTN1 impairs the clearance of stalled ribosomes at the ER (as shown in Fig. 2D,E), leading to the accumulation of UFMylated 60S-peptidyl-tRNA complex intermediates. Surprisingly, NEMF and LTN1 depletion did not significantly affect RPL26 UFMylation upon ANS treatment (Figs. 3C, lanes 5, 6 and 3D). Compared to steady-state conditions in Fig. 3A, this difference may be due to cell-wide ribosomal stalling induced by ANS treatment, which exhausts the UFMylation machinery. These data suggest that during ANS treatment, the UFMylation machinery operates at capacity; therefore, the depletion of NEMF and LTN1 does not result in a further increase under these conditions.

To dissect these mechanisms in experimental settings differentiating cytosolic and stalled ribosomes at the ER, we tested the effect of their loss in cells expressing the cytosolic or ER stalling reporters ERK20 and CytoK20. As shown before (Wang et al, 2020), while ERK20 expression increased RPL26 UFMylation, CytoK20 did not (Fig. 3E). The siRNA-mediated knockdowns of ZNF598 and ASCC3 resulted in a significant decrease in RPL26 UFMylation levels in cells expressing ERK20, compared to control siRNA, while PELO knockdown did not impact UFMylation (Figs. 3E, lanes 1–4 and 3F). In contrast, cells expressing CytoK20 showed no difference in RPL26-UFMylation levels upon knockdowns of ribosome rescue factors ZNF598 and ASCC3 (Fig. 3E, lanes 5–8 and 3G). Similarly, their depletion did not impact UFMylation levels in cells expressing control ERK0 reporter (Fig. EV3C,D). Therefore, the depletion of ZNF598 and ASC-1 complex decreases UFMylation, specifically in cells expressing ER stalling reporter ERK20. We also noticed an increase in UFMylation upon NEMF depletion in RKO iCas9 cells expressing the ERK20 reporter (Fig. 3H,I), indicating that UFMylation of RPL26 is not dependent on NEMF, consistent with our data showing that

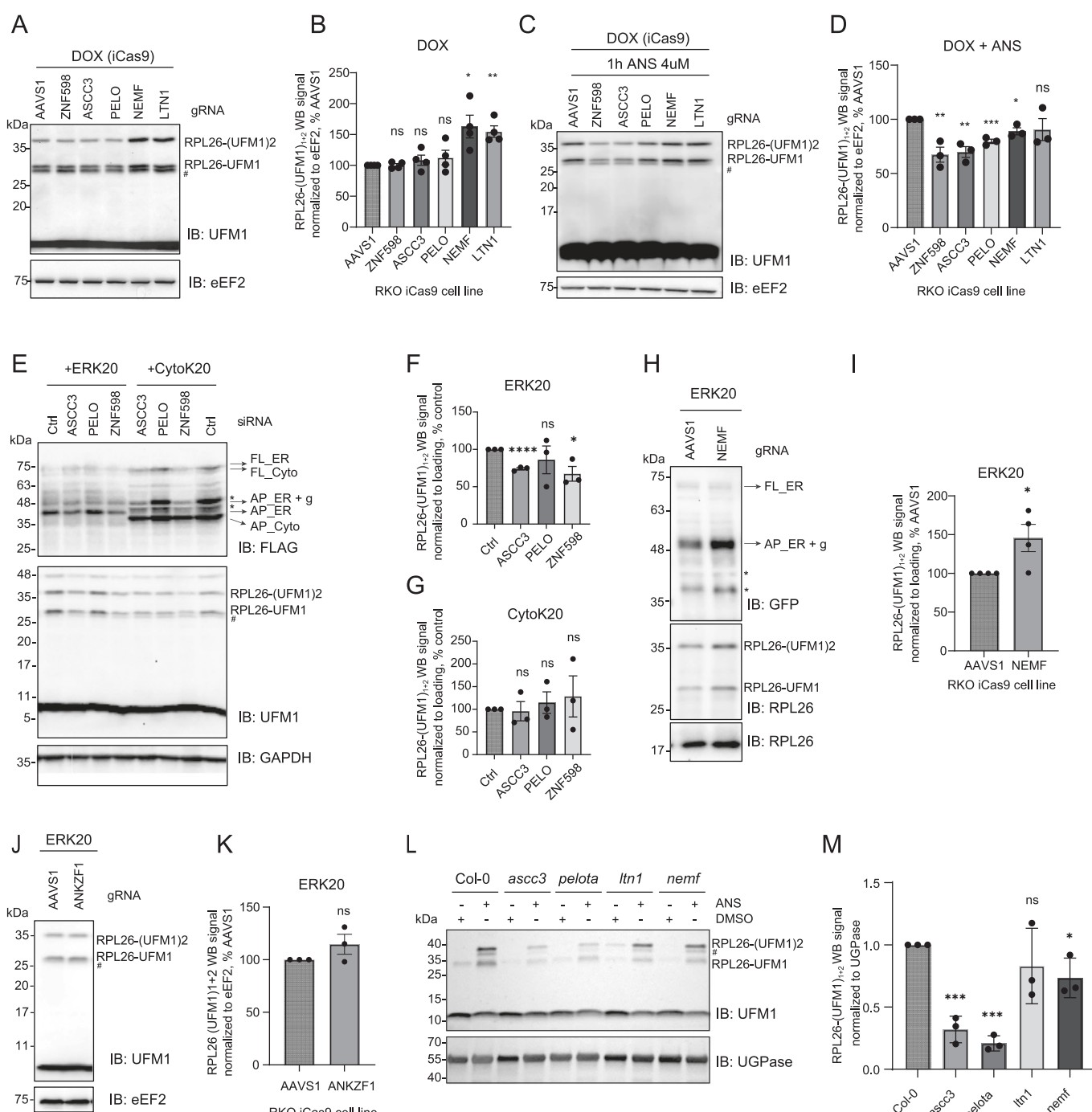

NEMF binds to already UFMylated 60S. However, we did not observe a significant increase in RPL26-UFMylation upon ANKZF1 depletion (Fig. 3J,K). This is in contrast to the increased stability of the ERK20 stalling reporter upon ANKZF1 depletion (Fig. 2D), suggesting that deUFMylation of RPL26 might precede ANKZF1-mediated hydrolysis of the peptidyl-tRNA bond.

Given the conservation of the UFMylation system across higher eukaryotes, we next investigated its evolutionary conservation in plants. To this end, we evaluated the effect of RQC machinery on RPL26 UFMylation using *Arabidopsis thaliana* mutant lines.

UFMylation levels were assessed upon ANS treatment in *ascc3*, *pelota*, *ltn1*, and *nemf* mutant lines compared to wt (Columbia Col-0 ecotype). Knocking out *ascc3* and *pelota* significantly decreased UFMylation levels upon ANS-induced stalling compared to the wild-type (Fig. 3L,M). Additionally, we noticed a slight decrease in the *nemf* mutant line (Fig. 3L,M). These results confirm our hypothesis that the order of events upon ribosome stalling at the ER is conserved across eukaryotes. Interestingly, in plants, we observed a more substantial effect of Pelota depletion on RPL26 UFMylation than in human cell lines. We speculate that differences in RNase

**Figure 3.  RPL26 is UFMylated on post-split 60S subunit upon ribosomal stalling at the ER.**

(A, C) UFMylation levels visualized by UFM1 immunoblot in RKO iCas9 cells upon 48 h dox treatment to induce RQC components or non-targeting AAVS1 knockout in untreated (A) or 1 h 4 μM ANS treated cells (C). (B) Quantification of (A) with unpaired two-sided Student t-test, n = 4. Error bars represent SEM. (D) Quantification of (C) with unpaired two-sided Student t-test, n = 3. Error bars represent SEM. (E) UFMylation levels visualized by UFM1 immunoblot in HCT116 cells upon 72 h siRNA-mediated knockdown of RQC components or non-targeting control, and 24 h expression of ERK20 or CytoK20. (F, G) Quantification of (E) for cells expressing ERK20 (F) or CytoK20 (G) shown by FLAG signal normalized to loading control with unpaired two-sided Student t-test, n = 3. Error bars represent SEM. (H) UFMylation levels visualized by RPL26 immunoblot in RKO iCas9 cells upon 48 h dox treatment to induce non-targeting AAVS1 or NEMF knockout upon 24 h expression of ERK20 reporter. (I) Quantification of (H) with unpaired two-sided Student t-test, n = 4. Error bars represent SEM. (J) UFMylation levels visualized by UFM1 immunoblot in RKO iCas9 cells upon 48 h dox treatment to induce non-targeting AAVS1 or ANKZF1 knockout upon 24 h expression of ERK20 reporter. (K) Quantification of (J) with unpaired two-sided Student t-test, n = 3. Error bars represent SEM. (L) Seven-day-old *Arabidopsis* seedlings of wild-type (Col-0) and RQC component mutant (*ascc3, pelota, ltn1, nemf, rqc1, edf1*) were treated with either DMSO or 100 μM ANS for 16 h. The UFMylation level was tested via immunoblotting using anti-UFM1 antibody, with UGPase was introduced as loading control. The data shown are representative of three biological replicates. (M) Quantification of (L) for ANS-treated samples with unpaired two-sided Student t-test, n = 3. Error bars represent standard deviation. ****P ≤ 0.0001, ***P ≤ 0.001, **P ≤ 0.01, *P ≤ 0.05. List of complete P values available in Dataset EV2. FL full length, AP arrested peptide, g glycosylation, # - UFC1-UFM1 complex (Kumar et al, 2021), *degradation products. Source data are available online for this figure.

activity in plants may create 3'-ends, inducing ribosome-stalling that are primarily resolved by Pelota (Yu et al, 2016; Tanaka et al, 2025).

Our data show that RPL26 UFMylation is diminished in the absence of either ZNF598 or ASC-1 complex, suggesting that recognition of collisions (mediated by ZNF598 (Juszkiewicz et al, 2018)) and subsequent ribosome splitting (mediated by ASC-1 complex (Juszkiewicz et al, 2020)) precede and are prerequisites for RPL26 UFMylation. At the same time, the depletion of the RQC components NEMF and LTN1, which are involved in the clearance of the arrested peptides on the 60S, does not impair the UFMylation of RPL26. These results indicate that the nascent polypeptide release is not necessary for the UFM1 E3 ligase complex to access and UFMylate the 60S. Taken together, we mechanistically show that translational stalling at the ER induces UFMylation of post-split 60S-peptidyl-tRNA complexes, and this mechanism is evolutionarily conserved in plants and mammals.

## UFM1 E3 ligase complex binds the nascent chain-associated 60S-peptidyl-tRNA complex at the ER

The cryo-electron microscopy structures of UFM1 E3 ligase complex bound to 60S ribosomes revealed that it forms extensive contacts with the 60S in a clamp-like architecture extending from tRNA-binding sites to the peptide exit tunnel, with all three subunits of the complex contributing to this interaction (DaRosa et al, 2024; Makhlouf et al, 2024). Based on these structures, it was proposed that UFL1 binding blocks the tRNA-binding site, and these two binding events are mutually exclusive. However, those structures cannot explain how the loss of the UFM1 E3 ligase components leads to the accumulation of ER-stalling reporters. Namely, it is not clear how the loss of machinery that only works on post-termination ribosomes could result in the accumulation of the arrested nascent polypeptides (Fig. 2D,E). To this end, we next dissected whether binding of the UFM1 E3 ligase complex occurs before or after nascent chain release from ER-bound 60S subunits.

Depletion of the RQC factors involved in the nascent chain clearance increased UFMylation of RPL26, suggesting that the UFM1 E3 ligase complex can access and UFMylate nascent chain-associated 60S subunits (Fig. 3A,B,L,M). The catalytic component of the UFM1 E3 ligase complex, UFL1, forms the central scaffold of the UFM1 E3 ligase complex and forms extensive contacts with the

other two components of the E3 ligase complex and the 60S (DaRosa et al, 2024; Makhlouf et al, 2024). The structural model of UFL1 displays a short N-terminal α-helix followed by one partial winged-helix (pWH), five WH motifs, a bipartite coiled-coil (CC) domain with a disordered region and a C-terminal globular domain (CTD), which contacts the 28S ribosomal RNA (rRNA), occluding all three tRNA-binding sites (Fig. 4A,B) (DaRosa et al, 2024; Makhlouf et al, 2024). Based on our data and data from others (DaRosa et al, 2024; Makhlouf et al, 2024), we hypothesized that the extensive contacts formed between the UFM1 E3 ligase complex and the 60S would allow for sufficient affinity for this interaction to occur even though the tRNA at the P-site is occupied. Notably, the structural models of the UFL1 show that CTD displays a high degree of conformational freedom due to it being connected to the rest of the protein with a disordered segment (Fig. 4A). Overlaying the published structures of 60S ribosomes with UFL1 complex and NEMF/LTN1 shows a possibility of a hybrid state with a possible rearrangement of the CTD of UFL1 and NEMF/LTN1 at the P-site (Fig. 4B). Supporting this, deletion of the CTD of UFL1 does not entirely abolish UFMylation of RPL26 in cells (DaRosa et al, 2024). Therefore, we speculated that UFL1 could undergo conformational rearrangements to enable its binding to the 60S-peptidyl-tRNA complex (Fig. 4B).

To experimentally test this model, we assessed whether the UFM1 E3 ligase complex associates with the 60S-peptidyl-tRNA complex. To this end, we enriched nascent chain-bound ribosomes by immunoprecipitating ER stalling reporter ERK20 or control ER reporter ERK0 in RKO cells (Fig. 4C). To enrich for the ribosome-associated nascent chains, we first isolated ribosomes via sucrose cushions and performed immunoprecipitation (Fig. 4C). We found that UFL1 specifically associates with ERK20 on nascent chain-bound ribosomes, whereas we did not observe UFL1 binding to ERK0 (Fig. 4D). Quantification of three independent replicates showed a significant difference in binding between ERK0 and ERK20, showing that UFL1 binding depends on ribosome stalling at the ER (Fig. 4E). Importantly, we also observed an enrichment of the UFMylated RPL26 in the eluates from ERK20-expressing cells (Fig. 4D, lanes 3–5). These data suggest that the UFM1 E3 ligase complex can bind and actively UFMylate the 60S-peptidyl-tRNA complex with arrested nascent chains at the ER. Similarly, as expected, we found that under those conditions, NEMF specifically associated with ERK20-containing ribosomes, validating our biochemical approach (Fig. 4F,G).

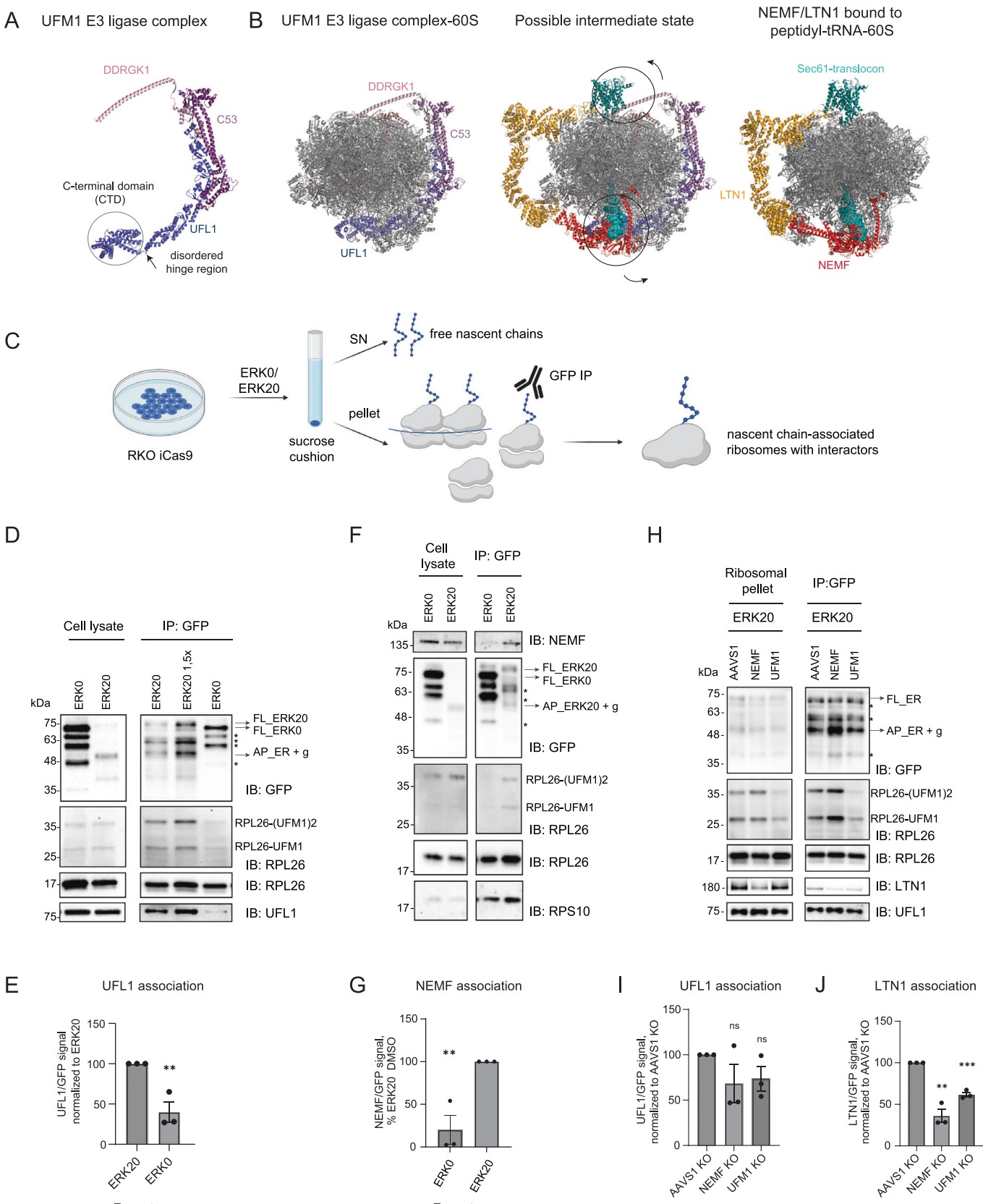

**Figure 4.   UFL1 UFMylates nascent chain-bound 60S subunits.**

(A) The cryostructural model of UFM1 E3 ligase complexes shows flexibility around the CTD region of UFL1 (pbd: 8ohd). (B) Left: The structure of UFM1 E3 ligase complex bound to 60S subunit (pbd: 8ohd), right: the structure of NEMF/LTN1 complex bound to peptidyl-tRNA-60S complex (pdb: 3j92, 8agw) docked on translocon (pdb:3j7r), middle: overlay of two structures shows a possible hybrid state with conformational rearrangements of UFL1 and DDRGK1. (C) Experimental approach for detecting interactors of nascent chain-associated ribosomes. (D) Ribosomal pellet was obtained by sucrose cushion centrifugation followed by GFP immunoprecipitation from RKO iCas9 parental cell line upon 24 h expression of ERK0 or ERK20 reporter. Protein levels are analyzed by immunoblotting. (E) Quantification of UFL1 association normalized to eluate nascent chain levels from (D) with unpaired two-sided Student $t$-test, $n = 3$. Error bars represent SEM. (F) Ribosomal pellet was obtained by sucrose cushion centrifugation followed by GFP immunoprecipitation from RKO iCas9 AAVS1 (non-targeting control) cell line upon 24 h expression of ERK0 or ERK20 reporter. Protein levels are analyzed by immunoblotting. (G) Quantification of NEMF association normalized to eluate nascent chain levels from (I) with unpaired two-sided Student $t$-test, $n = 3$. Error bars represent SEM. (H) Ribosomal pellet was obtained by sucrose cushion centrifugation followed by GFP immunoprecipitation from RKO iCas9 AAVS1 (non-targeting control), NEMF or UFM1 KO cell line upon 24 h expression of ERK20 reporter. Protein levels are analyzed by immunoblotting. (I) Quantification of UFL1 association normalized to eluate nascent chain levels from (H) with unpaired two-sided Student $t$-test, $n = 3$. Error bars represent SEM. (J) Quantification of LTN1 association normalized to eluate nascent chain levels from (H) with unpaired two-sided Student $t$-test, $n = 3$. Error bars represent SEM. ****$P \leq 0.0001$, ***$P \leq 0.001$, **$P \leq 0.01$, *$P \leq 0.05$. List of complete $P$ values available in Dataset EV2. FL full length, AP arrested peptide, # - UFC1-UFM1 complex (Kumar et al, 2021), * degradation products. Source data are available online for this figure.

To assess whether the association of NEMF/LTN1 with the 60S is required for UFL1 binding to the 60S-peptidyl-tRNA complex at the ER, we conducted similar experiments upon NEMF depletion in RKO iCas9 cells expressing ERK20 reporter (Fig. 4C). The IP analyses showed that UFL1 binding was unaffected by NEMF depletion, supporting the evidence that UFMylation precedes the activity of the late RQC components (Fig. 4H,I). Notably, upon NEMF depletion, EK20 pulldowns displayed higher levels of UFMylated RPL26, further supporting our model that the UFM1 E3 ligase complex binds to and UFMylates the 60S-peptidyl-tRNA complex before nascent chain release (Fig. 4H, lanes 4, 5). The UFM1 depletion did not impair the association of UFL1 with ERK20-expressing ribosomes, indicating that UFMylation of RPL26 does not largely stabilize UFL1 binding (Figs. 4H, lanes 4, 6 and 4I). We also noticed a NEMF-dependent association of LTN1 to the 60S-peptidyl-tRNA complexes for ER-stalled ribosomes (Fig. 4H,J), in accordance with previous data shown for cytosolic stalled ribosomes (Shao et al, 2015). Interestingly, we discovered that LTN1 binding decreases upon UFM1 KO for ER-stalled ribosomes (Fig. 4H,J). These data indicate that the UFM1 E3 ligase complex facilitates the binding of RQC components to the 60S peptidyl-tRNA at the ER. We envision that, as shown for the empty 60S-translocon interaction (DaRosa et al, 2024; Makhlouf et al, 2024), the UFMylation machinery destabilizes the translocon-60S interface. This would facilitate LTN1's access to the nascent chain and would allow positioning of LTN1's C-terminus, which contains the RING domain, in its canonical binding site near the exit tunnel. This thereby stabilizes LTN1's interaction with the complex (Fig. 4B).

This model predicts that impairment of RPL26 UFMylation would reduce LTN1's access to nascent chains and diminish LTN1-mediated ubiquitination of arrested polypeptides. To test whether impairing UFMylation affects LTN1-mediated ubiquitination of arrested polypeptides, we used tandem ubiquitin-binding entity (TUBE) pulldowns under control conditions and upon UFM1 or LTN1 depletion. In these assays, we expressed either the ERK0 control or the ERK20 stalling reporter, then enriched for ubiquitinated proteins using TUBEs. Surprisingly, those assays showed no consistent decrease in ERK20 ubiquitination upon depletion of either UFM1 or LTN1, under both steady-state conditions and in the presence of proteasome inhibitors (Figs. 5A,B and EV4A–D). Given the clear accumulation of ERK20 upon depletion of LTN1 and UFM1, we speculate that other ER-localized

E3 ligases might ubiquitinate the ER stalling reporter when canonical clearance mechanisms are impaired, thereby masking the effects of LTN1 or the UFM1 machinery. An unbiased genome-wide genetic screen that used accumulation of an ER-stalling reporter as a readout identified LTN1 as one of the top hits but did not identify any other well-characterized ER-localized E3 ligases (Wang et al, 2023). Similarly, depletion of different E3 ligases involved in ER-associated degradation (ERAD) does not stabilize ER-stalling reporters in either mammals or yeast (Scavone et al, 2023; Crowder et al, 2015). Altogether, these data converge on the model that while other E3 ligases might ubiquitinate stalled polypeptides at the ER, the RQC is the primary path for nascent chain degradation upon ribosomal stalling at the ER.

## DeUFMylation of RPL26 is not required for LTN1 binding

Recent structures of the UFM1 E3 ligase in complex with empty 60S suggested that upon UFMylation of RPL26, the UFM1 E3 ligase complex associates more stably with the 60S and deUFMylation of RPL26 is required for the disassembly of the UFM1 E3 ligase from the 60S ribosomal subunit (DaRosa et al, 2024; Makhlouf et al, 2024). We wondered whether deUFMylation is necessary for LTN1 association with the 60S-peptidyl-tRNA complex. To test this, we compared association of LTN1 with 60S-peptidyl-tRNA complexes under control conditions and upon depletion of deUFMylase UFSP2 using nascent chain pulldowns. To avoid adaptation of cells to long-term depletion of UFSP2, we used the iCas9 system. As expected, depletion of UFSP2 led to increased levels of UFMylated RPL26 (Fig. 5C). However, we did not observe a significant impact of UFSP2 depletion on LTN1 association with the 60S-peptidyl-tRNA complex, indicating that deUFMylation is not necessary for LTN1 association (Fig. 5C–E).

In summary, our data suggest that the UFM1 E3 ligase complex binds to the 60S-peptidyl-tRNA complexes after ribosome splitting at the ER. The spatial proximity of the ER-tethered UFM1 E3 ligase allows it to access the post-split ribosomes at the ER to perform UFMylation. This interaction does not depend on the binding of RQC components (NEMF/LTN1 complex), suggesting that it can precede RQC binding. Instead, RPL26 UFMylation facilitates the subsequent association of the RQC machinery with 60S-peptidyl-tRNA complexes, allowing the clearance of arrested polypeptides on ER-stalled ribosomes.

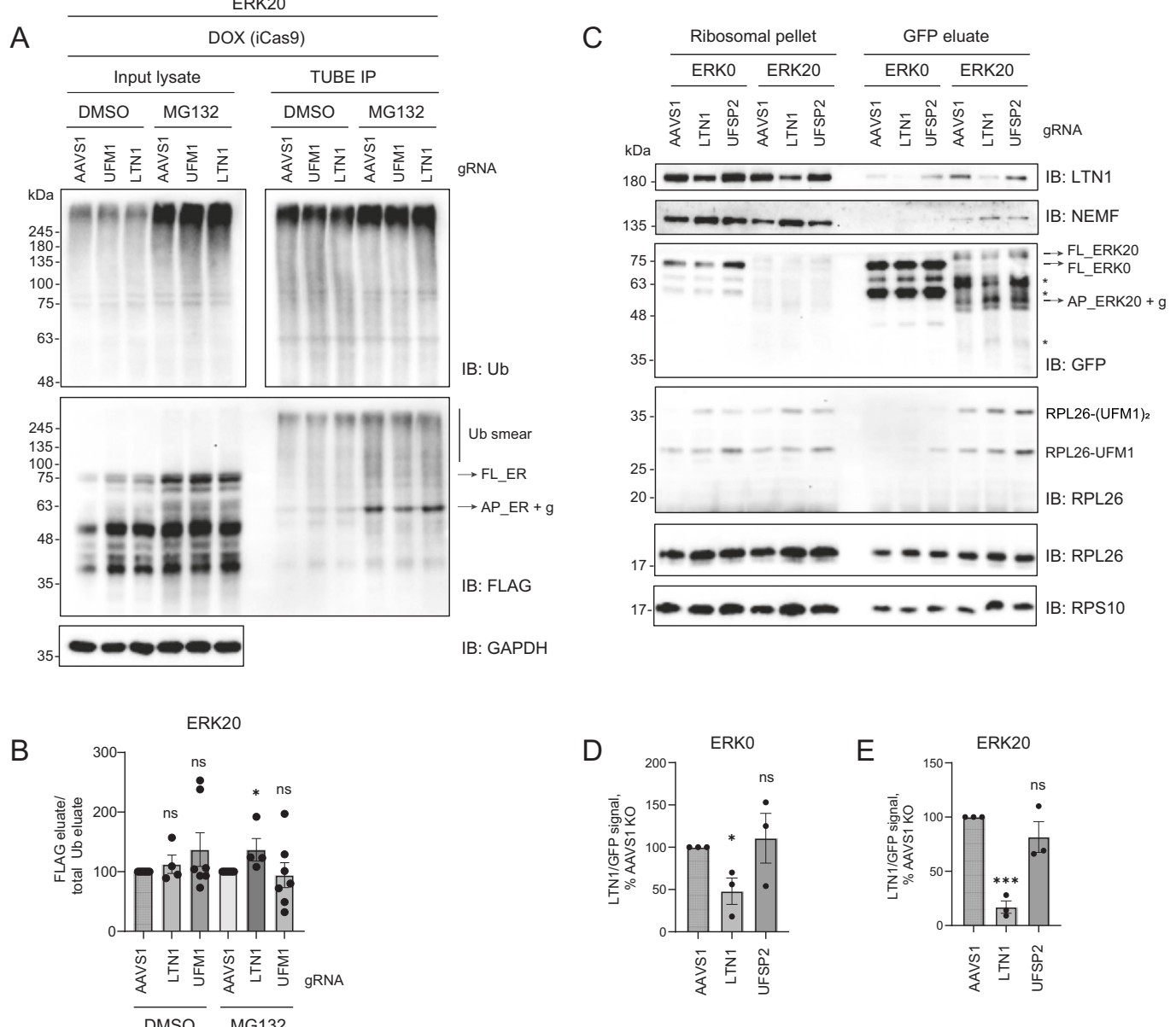

**Figure 5. UFSP2 does not impact LTN1 binding to 60S-peptidyl-tRNA complexes.**

(A) TUBE assays were performed from RKO iCas9 AAVS1 (non-targeting control), LTN1 or UFM1 KO cell line upon 24 h expression of ERK20 reporter and 3 h treatment with 20 μM proteasome inhibitor MG132 or DMSO as control. Protein levels are analyzed by immunoblotting. (B) Quantification of (A) with unpaired two-sided Student *t*-test, *n* = 7. Error bars represent SEM. (C) Ribosomal pellet was obtained by sucrose cushion centrifugation followed by GFP immunoprecipitation from RKO iCas9 AAVS1 (non-targeting control), LTN1 or UFSP2 KO cell line upon 24 h expression of ERK0 or ERK20 reporter. Protein levels are analyzed by immunoblotting. (D, E) Quantification of LTN1 association normalized to eluate nascent chain levels from (C) for cells expressing ERK0 (D), or ERK20 (E) with unpaired two-sided Student *t*-test, *n* = 3. Error bars represent SEM ****$P \le 0.0001$, ***$P \le 0.001$, **$P \le 0.01$, *$P \le 0.05$. List of complete *P* values available in Dataset EV2. FL full length, AP arrested peptide, g glycosylation, * degradation products. Source data are available online for this figure.

## Discussion

The best-described substrate of the UFMylation machinery is the large ribosomal subunit protein RPL26, initially discovered to be increasingly UFMylated upon ribosomal stalling at the ER (Wang et al, 2020; Walczak et al, 2019). However, the primary function of this event has remained poorly understood. Recent cryo-EM structures uncovered a novel function of the UFMylation machinery in recycling the 60S subunits after translational termination at the ER by releasing them from the translocon (DaRosa et al, 2024; Makhlouf et al, 2024). These structures also showed that UFL1 occupies the tRNA binding sites at the 60S, suggesting that the UFM1 E3 ligase binds to 60S after translation termination (DaRosa et al, 2024; Makhlouf et al, 2024). Therefore, the role of UFMylation in clearing the nascent polypeptides arrested at the ER-stalled ribosomes remained unclear.

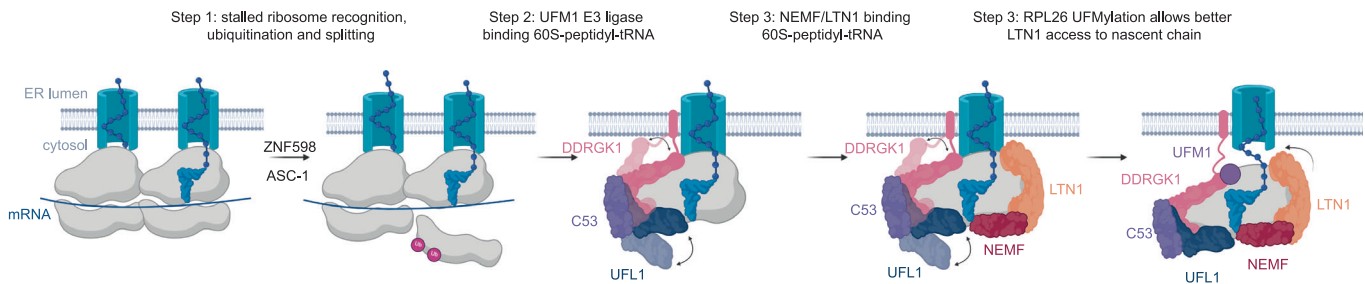

**Figure 6.   UFMylation machinery acts together with RQC to clear arrested peptides at the ER.**

Upon ribosomal stalling at the ER, ZNF598 recognizes and ubiquitinates 40S proteins, followed by ASC-1 complex-mediated splitting of the leading ribosome. The 60S-peptidyl-tRNA complex is recognized and UFMylated by the UFM1 E3 ligase complex, independently of downstream RQC factors. NEMF binds UFMylated 60S-peptidyl-tRNA complexes and brings in LTN1. UFMylation-induced translocon release from 60S-peptidyl-tRNA complexes allows LTN1 to access the nascent chain. LTN1 then ubiquitinates the arrested nascent chains. The nascent chain is released and subsequently degraded by the proteasome, while the 60S subunit is recycled.

Here, using IP analyses coupled to MS and Western blotting, we found that the RQC components are associated with the UFMylated ribosomes, indicating a direct physical link between these two pathways (Fig. 1A–E). To confirm the role of RQC factors in clearing the nascent polypeptides arrested at the ER-stalled ribosomes, we used genetic depletion of ribosome rescue and RQC machinery and found that ribosomes stalled upon ERK20 expression recruit the RQC E3 ligase ZNF598 and the ribosome-splitting factor ASC-1 complex (Figs. 2B,C and 3E,F). Similar to RQC events upon cytosolic stalling, the stalled ribosomes at the ER required the activity of ANKZF1, NEMF and LTN1 for clearance of arrested polypeptides since the loss of these components stabilized the ER stalling reporter (Fig. 2D,E and 3H,I). Likewise, impaired UFMylation specifically stabilized ERK20 substrate, in line with the published work (Fig. 2D,E) (Wang et al, 2020), indicating that both the RQC and the UFMylation machinery are involved in the clearance of the arrested polypeptides at the ER.

After demonstrating the role of the RQC and the UFMylation machinery in the clearance of the nascent chains arrested at the ER-stalled ribosomes, we assessed the sequence of events and interdependence of these pathways by testing the impact of the loss of ribosome rescue and RQC factors on RPL26 UFMylation upon ribosome stalling. The RPL26 UFMylation levels decreased upon knockdown or knockout of ZNF598 or ASCC3 in cells treated with anisomycin, a drug inducing ribosomal stalling (Figs. 3C,D and EV3A,B) as well as upon expression of the specific ER stalling reporter ERK20 (Fig. 3E,F). Experiments performed in plants (*Arabidopsis thaliana*) validated those findings (Fig. 3L,M), confirming that UFMylation happens on post-split 60S ribosomes upon ribosomal stalling at the ER and that this mechanism is conserved from plants to mammals.

Previous structural data showed that UFL1 binds to the 60S ribosomal subunit in a way that clashes with the binding of the 40S subunit, translocon, or tRNA (DaRosa et al, 2024; Makhlouf et al, 2024). These data cannot be reconciled with the data showing increased UFMylation of RPL26 upon ribosome stalling. We showed that UFL1 associates with 60S subunits upon translation of the ER stalling reporter ERK20, but not the ER-targeted control reporter ERK0 (Fig. 4D,E). Immunoprecipitation of the ribosome-attached ERK20 stalling reporter showed that these ribosomes were already UFMylated at RPL26, demonstrating that UFM1 E3 ligase

can act on nascent chain-loaded ribosomes with occupied tRNA at the P-site, contradicting the previous model (Fig. 4D,E) (DaRosa et al, 2024; Makhlouf et al, 2024). Supporting our findings, single particle cryo-electron microscopy studies of the native UFM1 E3 ligase complexes isolated from cells showed a small population of 60S with a weak extra density in the peptide exit tunnel, which could represent a nascent polypeptide chain (DaRosa et al, 2024). Importantly, the depletion of NEMF did not impact UFL1 binding to the nascent chain-containing 60S-peptidyl-tRNA complexes (Fig. 4H,I). In line with those results, we readily detected UFMylated RPL26 in the nascent chain immunoprecipitates from NEMF KO cells, showing that UFMylation does not depend on NEMF activity. Instead, NEMF depletion led to the accumulation of stalled ribosomes at the ER, observed by increased RPL26 UFMylation (Fig. 4H). These results indicated that UFMylation occurs before the clearance of the arrested polypeptides. To sum up, we found that the 60S-peptidyl-tRNA complex formed upon ribosome splitting is a canonical substrate for UFM1 E3 ligase complex and that the NEMF/LTN1-dependent release of the nascent chain is not necessary for RPL26 UFMylation. Instead, NEMF/LTN1 binds to the UFMylated ribosomes to clear arrested nascent chains and complete the ribosome rescue and RQC cycle. Supporting our findings, during the revision of our manuscript, a single-particle cryo-EM structure was published, demonstrating that the UFM1 E3 ligase complex associates with NEMF and LTN1 (Penchev et al, 2025). Notably, the structure revealed that the CTD of UFL1 adopts a novel conformation to allow its binding to the 60S-peptidyl-tRNA complex and NEMF (Penchev et al, 2025). Our data and those of others show that cells evolved intricate coordination between two distinct quality control pathways to enable efficient clearance of stalled polypeptides at the ER.

On cytosolic ribosomes, LTN1 binds close to the ribosome exit tunnel in proximity to the stalled nascent polypeptides emerging from the ribosome (Shao et al, 2015). On the ER-bound ribosomes, the exit tunnel is occupied by the translocon (Lewis et al, 2024; Voorhees et al, 2014). The LTN1 was proposed to access arrested polypeptides via their exposure to the cytosol by a gap at the ribosome-translocon contact site (von der Malsburg et al, 2015). As impaired UFMylation stabilizes the arrested nascent chains (Fig. 2D,E), and diminishes the binding of LTN1 to 60S at the ER (Fig. 4H,J), we speculate that, similar to what has been proposed

for the post-termination empty 60S subunits (DaRosa et al, 2024; Makhlouf et al, 2024), binding of the UFM1 E3 ligase complex destabilizes the 60S-translocon interactions and allows the E3 ligase LTN1 to access the arrested nascent chain for ubiquitination and subsequent degradation (Fig. 4B).

Both the ubiquitin-proteosome system and lysosomal degradation were proposed to be involved in the clearance of stalled nascent peptides at the ER (Wang et al, 2020; Stephani et al, 2020; Scavone et al, 2023). Notably, the previous work on the ERK20 reporter used here showed that it is mainly degraded by lysosomes and neither proteosome inhibition nor NEMF depletion stabilized the reporter (Wang et al, 2020). However, we show a clear stabilization of the same reporter preferentially by proteosome inhibition (Fig. EV2D), and NEMF depletion (Fig. 2D,E). Supporting our results, an alternative ER stalling reporter containing a folded VHP domain and a poly-lysine stretch was degraded primarily by the proteasome and showed NEMF-dependent stabilization (Scavone et al, 2023). We therefore conclude that while the RQC and ubiquitin-proteasome system play the major role in clearing stalled ribosomes, depending on the cell type and expression levels, arrested polypeptides at the ER can be cleared by complementary degradation pathways, including ER-phagy and lysosomal pathways (Inada and Beckmann, 2024; Bengtson and Joazeiro, 2010; Wang et al, 2020; Stephani et al, 2020).

Altogether, our data converge on the following model: stalling of the ER-bound ribosomes recruits canonical ribosome rescue and ribosome-associated quality control machinery. First, the E3 ligase ZNF598 recognizes the collided ribosomes and stabilizes them by ubiquitinating small subunit proteins (Juszkiewicz et al, 2018). This is followed by the binding of the ASC-1 complex that splits the leading ribosome, leaving a 60S-peptidyl-tRNA complex (Fig. 6). The 60S-peptidyl-tRNA complex is then recognized and UFMylated by the UFM1 E3 ligase complex. The association of the UFM1 E3 ligase complex destabilizes 60S-peptidyl-tRNA-translocon interactions, allowing the RQC components NEMF and LTN1 to bind and access the nascent chain, which is ubiquitinated and targeted for downstream proteasomal degradation to restore cellular proteostasis. This process shows high evolutionary conservation in plants and mammals. Our work highlights a fine collaboration between two distinct processes to maintain translational fidelity by overcoming the unique challenges posed by the spatial separation of arrested nascent chains from the ribosome at the ER.

# Methods

### Reagents and tools table

| Reagent/resource | Reference or source | Identifier or catalog number |
|---|---|---|
| **Experimental models** | | |
| RKO-Dox-Cas9 (iCas9) | Hornegger et al, 2024, de Almeida et al, 2021 | |
| RKO-Dox-Cas9 (iCas9) 3xFLAG-UFM1 | This study | |
| RKO-Dox-Cas9 (iCas9) PELO KO | This study | |

| Reagent/resource | Reference or source | Identifier or catalog number |
|---|---|---|
| RKO-Dox-Cas9 (iCas9) LTN1 KO | This study | |
| RKO-Dox-Cas9 (iCas9) NEMF KO | This study | |
| RKO-Dox-Cas9 (iCas9) ASCC3 KO | This study | |
| RKO-Dox-Cas9 (iCas9) ZNF598 KO | This study | |
| RKO-Dox-Cas9 (iCas9) UFM1 KO | This study | |
| RKO-Dox-Cas9 (iCas9) ANKFZ1 KO | This study | |
| RKO-Dox-Cas9 (iCas9) UFSP2 KO | This study | |
| HCT116 tetON OsTIR1 | Natsume et al, 2016 | |
| HCT116 tetON OsTIR1 3xFLAG-UFM1 | This study | |
| HEK293T cells | Shotaro Otsuka lab | CRL-11268 (ATCC) |
| HEK293T splitGFP parental | Leonetti et al, 2016 | |
| HEK293T Sec61β splitGFP | Leonetti et al, 2016 | |
| ascc3 mutant (A. thaliana) | NASC | AT5G61140 |
| pelota mutant (A. thaliana) | NASC | AT4G27650 |
| nemf mutant (A. thaliana) | NASC | AT5G49930 |
| ltn1 mutant (A. thaliana) | NASC | AT5G58410 |
| **Recombinant DNA** | | |
| Hygro-P2A-mAID-mClover | Addgene | #121179 |
| BSD-P2A-3xFLAG HDR | This study | |
| Hygro-P2A-3xFLAG HDR | This study | |
| Dual-sgRNA_hU6-mU6 | de Almeida et al, 2021 | |
| pSpCas9 (BB)-2A-GFP (PX458) | Addgene | Cat #48138 |
| 6His-TEVb-HaloTR-TUBE | MRC PPU Dundee | Cat #DU23799 |
| CytoK0 | Wang et al, 2020 | |
| CytoK20 | Wang et al, 2020 | |
| ERK0 | Wang et al, 2020 | |
| ERK20 | Wang et al, 2020 | |
| **Antibodies** | | |
| Rabbit anti-ANKZF1 | Proteintech | 20447-1-AP |
| Rabbit anti-ASCC3 | Bethyl laboratories | A304-014A-T |
| Rabbit anti-DDRGK1 | Proteintech | 21445-1-AP |
| Rabbit anti-eEF2 | Proteintech | 20107-1-AP |
| Mouse anti-FLAG | Sigma | F1804-5MG |

| Reagent/resource | Reference or source | Identifier or catalog number |
|---|---|---|
| Rabbit anti-GAPDH | Proteintech | 10494-1-AP |
| Mouse anti-GFP | Roche | 11814460001 |
| Rabbit anti-LTN1 | Proteintech | 28452-1-AP |
| Rabbit anti-NEMF | Proteintech | 11840-1-AP |
| Rabbit anti-Pelota | Proteintech | 10582-1-AP |
| Rabbit anti-RPL26 | Abcam | ab59567 |
| Rabbit anti-RPS10 | Abcam | ab151550 |
| Rabbit anti-RPS20 | Abcam | ab133776 [EPR8716] |
| Rabbit anti-RPS3 | Proteintech | 11990-1-AP |
| Mouse anti-Sec61α | Santa Cruz | sc-393182 sample |
| Mouse anti-Ub | Santa Cruz | sc-8017 |
| Rabbit anti-UFL1 | Proteintech | 26087-1-AP |
| Rabbit anti-UFM1 | Abcam | ab109305 |
| Rabbit anti-UGPase | Agrisera | AS14 2813 |
| Mouse anti-Vinculin | Proteintech | 6305-1-Ig |
| Rabbit anti-ZNF598 | Sigma-Aldrich | HPA041760-25UL |
| **Oligonucleotides and other sequence-based reagents** | | |
| Single guide RNAs | This study | Dataset EV2 |
| Primers | This study | Dataset EV2 |
| ASCC3 siRNA | Dharmacon | L-012757-01-0005 |
| PELO siRNA | Dharmacon | L-019068-01-0005 |
| ZNF598 siRNA | Dharmacon | L-007104-00-0005 |
| Control siRNA | Dharmacon | D-001810-10-05 |
| **Chemicals, enzymes and other reagents** | | |
| digitonin | Carl Roth | 4005.4 |
| Anisomycin | Sigma-Aldrich | A9789-5MG |
| Doxycycline | Sigma-Aldrich | D5207-5G |
| MG132 | Santa Cruz Biotechnology | sc-201270 |
| Bafilomycin A1 | Szabo Scandic | AVIOOMC00028-1 |
| **Software** | | |
| GraphPad Prism 10 | https://www.graphpad.com/ | |
| Adobe Illustrator 2023 | https://www.adobe.com/products/illustrator.html | |
| BioRender | https://www.biorender.com/ | |
| FlowJo v10.9.0 | https://www.flowjo.com/ | |
| Pymol v 3.1.5.1 | https://www.pymol.org/ | |
| **Other** | | |
| Dynabeads™ Protein G | Invitrogen | 10004D |
| GFP-Trap® Magnetic Particles | ChromoTek | gtma-20 |
| Magne® HaloTag® Beads | Promega | G7281 |

## Mammalian cell culture

HCT116 tetON (doxycycline-inducible) OsTIR1 cells obtained the Masato Kanemaki lab (Natsume et al, 2016) were cultured in McCoy's 5 A (modified) medium (Sigma, M9309) supplemented with 10% fetal bovine serum (Gibco, 10437028), 2 mM L-Glutamine (Sigma, G7513), 1% Pen/Step (Sigma, P0781). RKO-Dox-Cas9 cells (expressing doxycycline-inducible Cas9), a kind gift from from Johannes Zuber lab (de Almeida et al, 2021) were cultured in RPMI-1640 media (Sigma, R8758) supplemented with 10% fetal bovine serum (Gibco, 10437028), 2 mM L-Glutamine (Sigma, G7513), 1% Pen/Step (Sigma, P0781), 1× non-essential amino acids (Thermo Scientific, 11140050), and 1 mM sodium pyruvate (Gibco, 11360070). HEK293T cells, a kind gift from Shotaro Otsuka lab, were cultured in DMEM medium (Sigma, D5796) supplemented with 10% fetal bovine serum (Gibco, 10437028), 2 mM L-Glutamine (Sigma, G7513), 1% Pen/Step (Sigma, P0781). Cell lines were grown in a humidified incubator at 37 °C and 5% $CO_2$. All cell lines were regularly tested for *Mycoplasma* infection with the EZ-PCR™ Mycoplasma Detection Kit (Biological Industries).

## Mammalian cell line generation

RKO-Dox-Cas9 (iCas9) cell lines for doxycycline inducible knock-out of the genes encoding RQC components were established using lentiviral transduction with lentiviral particles (produced as described in (Hornegger et al, 2024) containing Dual-sgRNA_hU6-mU6 vectors described in (de Almeida et al, 2021) expressing two sgRNAs (Dataset EV2) from human and mouse U6 promoters and eBFP2 from a PGK promoter. The eBFP2-positive cells were FACS sorted at BD FACSMelody™ Cell Sorter at Max Perutz Labs BioOptics FACS Facility.

To endogenously tag UFM1 with N-terminal 3xFLAG sequence human UFM1 N-terminal homology arms sequence was amplified using HindIII_UFM1_N_HAs_F and XhoI_UFM1_N_HAs_R primers. The UFM1 homology arms were inserted into the pMK344 plasmid backbone (Addgene #121179) using HindIII and XhoI restriction sites. To remove the BamHI site the plasmid was treated with HindIII and XbaI restriction enzymes and ligated through annealed overlapping oligos (pMK344_oligo_HindIII_XbaI_F/R). The BamHI and SalI restriction sites were introduced to the homology arms using BamHI_UFM1_N-HA and SalI_UFM1_N-HA primers. The BSD-P2A-3xFLAG sequence was cloned using the pMK347 (BSD-P2A-mAID) plasmid (Addgene #121181) which was digested with the Esp3I and BamHI enzymes to remove P2A-mAID and ligated using P2A-3xFLAG template made from annealed oligos (3xFLAG_pMK347_F/R) with cohesive ends. The BSD-P2A-3xFLAG sequence was inserted into the plasmid with N-terminal homology arms of UFM1 using SalI and BamI restriction sites resulting in the HDR template. The PAM sites were mutated in the HDR template using site-directed mutagenesis with a primer pair UFM1_gRNA_517_PAM_mut_F/R. To clone the Hygro-P2A-3xFLAG HDR template Hygro-P2A-3xFLAG was amplified from pMK344 with KS_F and P2A_3xFLAG_BamHI_R primers and cloned to replace the BSD-P2A-3xFLAG in the final UFM1 HDR template with mutated PAM sites. All primer sequences are listed in Dataset EV2.

To introduce the 3xFLAG tag to the N-terminus of the endogenous UFM1 HCT116 tetON OsTIR1 (Natsume et al, 2016) or RKO-Dox-Cas9 (iCas9) (Hornegger et al, 2024) cells were transiently transfected with 1:1:2 ratio mixture of the BSD-P2A-3xFLAG HDR, Hygro-P2A-3xFLAG HDR template plasmids and pSpCas9 (BB)-2A-GFP (PX458) (plasmid #48138, Addgene) (Ran et al, 2013) targeting the first exon of the UFM1 gene (Dataset EV2) using the Fugene HD (Promega) reagent according to the manufacturer's instructions. 24 h after transfection cell were collected by trypsinization and plated in 1:200 dilution in standard culture media (McCoy's 5 A (modified) with 10% FBS, 2 mM L-Glutamine, 1% Pen/Step). On the next day the media was supplemented with 100 μg/mL of Hygromycin B Gold (InvivoGen, #ant-hg) and 10 μg/mL of Blasticidin S Hydrochloride (InvivoGen, #ant-bl-05). Cells were grown in selection media until visible colonies were formed and expanded in 96-well plates. The homozygous insertion was verified using western blotting with anti-UFM1 (ab108062, Abcam) and anti-GAPDH (10494-1-AP, Proteintech) antibodies and genotyped using DirectPCR Lysis-Reagent Cell (Peqlab, VWR) with hsUFM1_HAs_F, hsUFM1_HAs_R, HygR_F, and BSDR_F primers.

## Co-immunoprecipitation mass spectrometry of 3xFLAG-UFM1

### Sample processing

For MS analysis of 3xFLAG-UFM1 co-immunoprecipitation (co-IP), four ⌀15 cm dishes of ~80% confluent 3xFLAG-UFM1 HCT116 cells per condition were used. Media was changed 16 h before collection. Ribosome stalling was induced with 200 nM anisomycin for 1 h. Cells were washed with warm PBS + 100 μg/mL cycloheximide (CHX), lysed on ice in buffer (20 mM HEPES pH 7.3, 120 mM KCl, 15 mM MgCl$_2$, 1.5% digitonin, 100 mg/mL CHX, 0.5 mM DTT, 1x protease inhibitor Roche), scraped, and incubated on ice with vortexing. Lysates were passed through a 27 G needle and centrifuged (20,000× g, 20 min, 4 °C). Clarified lysates (2.55 mL) were layered onto 850 μL of 25% sucrose cushions and centrifuged (100,000× g, 1.5 h, 4 °C, TLA100.3 rotor). Ribosomal pellets were resuspended in 1 mL of lysis buffer (20 mM HEPES pH 7.3, 120 mM KCl, 15 mM MgCl$_2$, 1.5% digitonin, 100 mg/mL CHX, 0.5 mM DTT, 1× protease inhibitor Roche). For co-IP, 40 μg anti-FLAG M2 antibody or IgG control were coupled to 160 μL protein G Dynabeads in 1 mL lysis buffer (without DTT/CHX), washed, and added to 450 μL resuspended ribosomes. After 2 h incubation (4 °C), beads were washed three times with (20 mM HEPES pH 7.3, 195 mM KCl, 5 mM MgCl$_2$, 0.5% digitonin, 100 mg/mL cycloheximide, 0.1 mM DTT, 1× EDTA-free protease inhibitor cocktail), followed by three washes with (20 mM HEPES pH 7.3, 195 mM KCl, 5 mM MgCl$_2$), then resuspended in 50 μL of 100 mM ammonium bicarbonate (ABC), supplemented with 300 ng trypsin (Trypsin Gold, Promega) and incubated for 4 h on a Thermo-shaker with 1200 rpm at 37 °C. The supernatant was transferred to a fresh tube and reduced with 0.5 mM Tris 2-carboxyethyl phosphine hydrochloride (TCEP, Sigma) for 30 min at 60 °C and alkylated in 3 mM methyl methanethiosulfonate (MMTS, Fluka) for 30 min at room temp protected from light. Subsequently, the sample was digested with another 300 ng of trypsin at 37 °C overnight. The digest was acidified by addition of trifluoroacetic acid (TFA, Pierce) to 1%. A similar aliquot of each sample was analyzed by LC-MS/MS. Total protein concentration of the lysate and ribosomal pellet inputs was measured using Pierce™ BCA Protein Assay Kit (Reducing Agent Compatible). Aliquots containing 25 μg of total protein were tryptic digested, using the iST kit (PO 00001, PreOmics) according to the manufacturer's instructions. Offline sample desalting using the OASIS MCX 96-well uElution Plate 30 μm (Waters), 2 μl sorbent per well, was performed in a centrifuge spinning at 200×g for 3 min at each step. Samples were brought to a concentration of 2% H3PO4 in 100 mM Ammonium Formate and loaded to dry wells of the OASIS plate. Bound peptides were washed 2× with 200 μl of 2% formic acid in 100 mM Ammonium Formate and 2 ×200 μl of 100% methanol. Elution of peptides was performed by applying 2×25 μl of 5% NH4OH in 80% methanol, followed by vacuum centrifugation to dryness and reconstitution in 20 μl of 2% acetonitrile in 0.1% TFA. 250 ng of each peptide sample each sample was analyzed by LC-MS/MS. The nano HPLC system (Vanquish Neo UHPLC-System, Thermo Scientific) was coupled to an Orbitrap Astral mass spectrometer, equipped with a Nanospray Flex ion source (all parts Thermo Scientific). Peptides were loaded onto a trap column (PepMap Acclaim C18, 5 mm × 300 μm ID, 5 μm particle size, 100 Å pore size, Thermo Scientific) at a flow rate of 25 μl/min using 0.1% TFA as mobile phase. After 5 min, the trap column was switched in line with the analytical column (Aurora Ultimate C18 25 cm × 75 μm ID, 1.7 μm particles, 120 Å, Ionopticks operated at 50 °C). Peptides were eluted using a flow rate of 300 nl/min, starting with the mobile phases 98% A (0.1% formic acid in water) and 2% B (80% acetonitrile, 0.1% formic acid) and linearly increasing to 35% B over the next 60 min, followed by an increase to 95% B in 1.7 min, a 4-min hold at 95% B, and re-equilibration with 2% B for three column volumes (equilibration factor of 3.0). The Orbitrap Astral was operated in data-dependent mode, performing a full scan in the Orbitrap every 0.7 s (m/z range 375–1500 m/z; resolution 240,000; standard AGC target, maximum injection time 10 ms, minimum intensity 5000). MS/MS spectra were acquired in the Astral analyser by isolating 1.6 Da windows and fragmenting precursor ions with normalized HCD collision energy of 30% with a maximum injection time of 5 or 10 ms correspondingly or until a standard AGC target was reached. Fragment ions ranging from 150 to 2000 m/z were acquired. Precursor ions selected for fragmentation (include charge state 2–6) were put on a dynamic exclusion list for 30 s.

### Data processing

For peptide identification from DDA data, the RAW-files were loaded into Proteome Discoverer (version 3.2.0.450, Thermo Scientific). All MS/MS spectra were searched using MSAmanda version 3.2.22.93 (Dorfer et al, 2014). The peptide and fragment mass tolerance was set to ±10 ppm, the maximum number of missed cleavages was set to 2, using tryptic enzymatic specificity without proline restriction. The RAW-files were searched against the uniprot reference database, using human as sub-organism (20,586 sequences; 11,456,034 residues), supplemented with common contaminants using the following modifications: Carbamidomethylation of cysteine was set as fixed modification and oxidation of methionine, UFMylation residues VG and DRVG each on Lysine, deamidation of asparagine and glutamine, glutamine to pyro-glutamate conversion at peptide N-terminal glutamine and acetylation on the protein N-terminus were set as variable

modifications. The localization of the post-translational modification sites within the peptides was performed with the tool ptmRS, based on the tool phosphoRS (Taus et al, 2011). The result was filtered to 1% FDR on PSM and protein level using the Percolator algorithm (Käll et al, 2007) integrated in Proteome Discoverer. Additionally, an Amanda score cut-off of at least 150 was applied. Proteins were filtered to be identified by a minimum of 2 PSMs in at least 1 sample. Protein areas were computed in IMP-apQuant (Doblmann et al, 2019) by summing up unique and razor peptides. Resulting protein areas were normalized using iBAQ (Schwanhäusser et al, 2011). Match-between-runs (MBR) was applied for peptides with high confident peak area that were identified by MS/MS spectra in at least one run. Proteins were filtered to be identified by a minimum of 3 quantified peptides. Statistical significance of differentially expressed proteins was determined using limma (Smyth, 2004).

## Mammalian siRNA and plasmid transfections

For reporter accumulation experiments, RKO iCas9 cells were seeded in 6-well plates and treated with 450 nM doxycycline for 48 h to induce expression of Cas9. The cells were then transfected with 2 µg plasmid using jetOPTIMUS® reagent according to the manufacturer's instructions (Avantor, 101000051). Cells were washed with cold PBS buffer and collected on plate in RIPA buffer (150 mM NaCl, 1% NP-40, 0.5% Sodium deoxycholate, 0.1% SDS, and 25 mM TRIS pH 7.4) with 1× EDTA-free protease inhibitor cocktail (Roche, 11873580001). Cell lysate was clarified on a tabletop centrifuge at maximum speed (20,000 g) for 20 min at 4 °C.

To test the effect of RQC components' loss on UFMylation, HCT116 tetON OsTIR1 cells were seeded in 24-well plates and transfected with Dharmacon ON-TARGETplus Human siRNA SMARTpools at 10 nM against ASCC3 (L-012757-01-0005) and at 25 nM against PELO (L-019068-01-0005) and ZNF598 (L-007104-00-0005). ON-TARGETplus Non-targeting Control Pool (D-001810-10-05) at 25 nM was used as a control. Cells were passed to 6-well plates and either transfected with 2 µg of plasmids containing CytoK0, CytoK20, ERK0, or ERK20 reporters (Wang et al, 2020) for 24 h using jetOPTIMUS® reagent (Avantor, 101000051) according to the manufacturer's instructions or treated with 4 µM anisomycin dissolved in DMSO for 1 h before collection.

RKO iCas9 cells were seeded in ∅10 cm dishes in media supplemented with 450 nM doxycycline for 48 h to induce expression of Cas9. Cells were then either transfected with 10 µg of plasmids containing CytoK0, CytoK20, ERK0, or ERK20 reporters (Wang et al, 2020) for 24 h using jetOPTIMUS® reagent according to the manufacturer's instructions (Avantor, 101000051), or treated with 4 µM anisomycin dissolved in DMSO for 1 h before collection.

## Flow cytometry analysis

HCT116 tetON OsTIR1 cells were seeded in 24-well plates and transfected with Dharmacon ON-TARGETplus Human siRNA SMARTpools for 72 h total as described in "Mammalian siRNA and plasmid transfections". Cells were passed to 6-well plates and transfected with 2 µg of pEGFP-N1 plasmid containing CytoK0, CytoK20, ERK0, or ERK20 reporters (Wang et al, 2020) for 24 h

using jetOPTIMUS® reagent according to the manufacturer's instructions (Avantor, 101000051). Cells were harvested in trypsin, resuspended in full media and analyzed by flow cytometry (BD LSRFortessa™ Cell Analyser). Data was analyzed in FlowJo software.

## Sucrose density gradient fractionation

For polysome sucrose density fractionation, four ∅15 cm dishes of ~80% confluent HCT116 3xFLAG-UFM1 cells were used per condition. Ribosome stalling was induced with 200 nM anisomycin for 1 h. Cells were washed in cold PBS supplemented with 100 µg/mL cycloheximide, lysed on the plate with 300 µL of ice-cold lysis buffer (20 mM HEPES pH 7.3, 120 mM KCl, 15 mM MgCl₂, 1.5% digitonin, 100 µg/mL cycloheximide, 0.5 mM DTT, 1× EDTA-free protease inhibitor cocktail). Cell lysates were further incubated on ice for 10 min with intermittent vortexing, passed three times through the 27 G needle, and clarified on a table-top centrifuge at maximum speed (20,000× g) for 10 min at 4 °C. In total, 1700 µL of the clarified lysate were layered on top of 10–35% sucrose gradient (gradient buffer composition: 20 mM HEPES pH 7.3, 120 mM KCl, 5 mM MgCl₂, 100 µg/mL cycloheximide, 0.5 mM DTT) and centrifugated at 35,000× g for 3 h at 4 °C using SW32Ti rotor. Polysome fractions were separated on a BioComp fractionator and 500 µL fractions were collected. Western blot analysis was performed following trichloroacetic acid (TCA) precipitation. Protein pellets were dissolved in 1× Leammli sample buffer. Fractions corresponding to the migration peaks of ribosomal subunits used for Western blotting were separated by at least four fractions.

For polysome sucrose density fractionation of HEK293T cells, one ∅15 cm dish at ~80% confluency was used per condition. The experiment was performed similarly to that in HCT116 3xFLAG-UFM1 cells, with the following modifications: 250 µL of clarified lysate were layered onto a 10–35% sucrose gradient and centrifuged using an SW40Ti rotor. Fractions of 200 µL were collected. Fractions corresponding to the migration peaks of ribosomal subunits used for Western blotting were separated by at least four fractions.

## Cell lysis and immunoblotting

Clear cell lysates were collected in RIPA buffer with 1× EDTA-free protease inhibitor cocktail (Roche, 11873580001), clarified on a table-top centrifuge at maximum speed (20,000× g) for 20 min at 4 °C, and analyzed on 10%, 12% or 15% SDS-polyacrylamide gels, followed by a semi-dry transfer (BioRad blotter) or wet transfer (120 V, 1 h 20 min on a BioRad blotter) on a 0.2 µm nitrocellulose membrane. Membrane was blocked in 1×TBS buffer with 0.1% Tween containing 5% milk for 1 h at room temperature. Primary antibody (listed in Dataset EV2) was diluted in 2.5% milk 1× TBS-0.1% Tween buffer and incubated either 1 h at room temperature or at 4 °C overnight, followed by three washes with 1× TBS-0.1% Tween buffer. Secondary antibody conjugated with horseradish peroxidase (Promega) was diluted in 2.5% milk 1× TBS-0.1% Tween buffer and incubated for 1 h at room temperature, followed by three washes with 2.5% milk 1× TBS-0.1% Tween buffer. The blots were developed using chemiluminescent detection on iBright ThermoFisher machine and quantified in iBright Analysis Software (ThermoFisher).

## Immunoprecipitations

### GFP IP from sucrose cushions

For detecting UFL1 and RQC factors binding to nascent chain-containing 60S ribosomes, RKO iCas9 cell lines expressing gRNA against AAVS1 (non-targeting control), NEMF, or UFM1 (gRNA sequences are listed in Dataset EV2) were seeded in ⌀10-cm dishes (2 dishes per condition) in media supplemented with 450 nM doxycycline for 48 h to induce expression of Cas9. Cells were then transfected with 10 μg of plasmids containing CytoK20, or ERK20 reporters (Wang et al, 2020) for 24 h using jetOPTIMUS® reagent according to the manufacturer's instructions (Avantor, 101000051). Cells were then washed in cold PBS supplemented with 100 μg/mL cycloheximide, lysed on the plate with ice-cold lysis buffer (20 mM HEPES pH 7.3, 120 mM KCl, 5 mM MgCl$_2$, 1% NP-40, 100 μg/mL cycloheximide, 0.5 mM DTT, 1× EDTA-free protease inhibitor cocktail). Cell lysates were further incubated on ice for 10 min with intermittent vortexing, passed three times through the 27 G needle, and clarified on a table-top centrifuge at maximum speed (20,000× g) for 20 min at 4 °C. 700 μL of the lysate was layered on top of 233 μL of sucrose cushion (1 M sucrose, 20 mM HEPES pH 7.3, 120 mM KCl, 5 mM MgCl$_2$, 100 μg/mL cycloheximide, 0.5 mM DTT). Ribosomes were pelleted by centrifugation at 100,000× g for 1.5 h at 4 °C in a TLA100.3 rotor. After the supernatant was removed, the ribosomal pellet was resuspended with 150 μL of lysis buffer. Immunoprecipitations were performed with 40 μL of resuspended GFP-Trap® Magnetic Particles (ChromoTek, gtd-20) per condition, the mixture was incubated on a rotator at +4 °C for 1 h, washed 3 times with wash buffer (20 mM HEPES pH 7.3, 195 mM KCl, 5 mM MgCl$_2$, 0.5% NP-40, 100 μg/mL cycloheximide, 0.1 mM DTT, 1× EDTA-free protease inhibitor cocktail), and eluted in 30 μL of 1× SDS sample buffer without DTT. Samples were incubated at 70 °C for 10 min, the eluate was separated from the magnetic particles, DTT was added to the final concentration of 10 mM, samples were incubated at 95 °C for 5 min and loaded on a SDS-PAGE gel followed by western blotting.

### FLAG IP from cell lysate

HCT116 tetON OsTIR1 3xFLAG-UFM1 or RKO iCas9 3xFLAG-UFM1 cell lines or their respective parental cell lines were seeded in ⌀15-cm dishes (2 dishes per condition) and treated with 200 nM ANS for 1 h before collection, or untreated as a control. Cells were washed with cold PBS and lysed on plate with ice-cold lysis buffer (20 mM HEPES pH 7.3, 120 mM KCl, 15 mM MgCl$_2$, 1.5% w/v digitonin, 100 μg/mL cycloheximide, 0.5 mM DTT, 1× EDTA-free protease inhibitor cocktail). Cell lysates were further incubated on ice for 10 min, passed three times through the 27 G needle, and clarified on a table-top centrifuge at maximum speed (20,000× g) for 20 min at 4 °C. For co-IP, 10 μg anti-FLAG M2 antibody was coupled to 40 μL protein G Dynabeads (per 15-cm dish) in 1 mL lysis buffer (without DTT/CHX) and coupled for 20 min at room temperature on an end-over-end rotator. Beads were washed three times with lysis buffer without DTT. Co-IP was performed from 700 μL of cleared cell lysates for 2 h at 4 °C on an end-over-end rotator. Beads were washed three times with the wash buffer (20 mM HEPES pH 7.3, 195 mM KCl, 5 mM MgCl$_2$, 0.5% w/v digitonin, 100 μg/mL cycloheximide, 0.1 mM DTT, 1× EDTA-free protease inhibitor cocktail) and eluted in 30 μL of 1× SDS sample buffer without DTT. Samples were incubated at 70 °C for 10 min,

the eluate was separated from the magnetic particles, DTT was added to the final concentration of 10 mM, samples were incubated at 95 °C for 5 min and loaded on a SDS-PAGE gel followed by western blotting.

### GFP IP from split GFP cells

HEK293 Sec61β split GFP or parental cell lines (Leonetti et al, 2016), a kind gift from Leonetti lab were seeded in 15-cm dishes (2 dishes per condition) and treated with 4 μM ANS for 3 h before collection, or untreated as control. Cells were washed with cold PBS and lysed on plate with ice-cold lysis buffer (20 mM HEPES pH 7.3, 150 mM KCl, 5 mM MgCl$_2$, 1,5% w/v digitonin, 0.5 mM DTT, 1× EDTA-free protease inhibitor cocktail). Cell lysates were further incubated on ice for 10 min, passed three times through the 27 G needle, and clarified on a table-top centrifuge at maximum speed (20,000× g) for 20 min at 4 °C. Cleared cell lysates were treated with 2.5 μg RNase A (Thermo Scientific, EN0531) per 750 ng total RNA for 20 min at 25 °C at 750 rpm. Immunoprecipitations were subsequently performed with 25 μL of resuspended GFP-Trap® Magnetic Particles (ChromoTek, gtma-20) per dish, the beads were incubated with 830 μL of cell lysate on a rotator at +4 °C for 2 h, washed three times with wash buffer (20 mM HEPES pH 7.3, 150 mM KCl, 5 mM MgCl$_2$, 0.5% w/v digitonin, 0.1 mM DTT, 1× EDTA-free protease inhibitor cocktail), and eluted in 25 μL of 1× SDS sample buffer with 10 mM DTT, boiled at 95 °C for 5 min and loaded on a SDS-PAGE gel followed by western blotting.

## TUBE assays

For detecting polyubiquitinated proteins, RKO iCas9 cell lines expressing gRNA against AAVS1 (non-targeting control), UFM1, or LTN1 (gRNA sequences are listed in Dataset EV2) were seeded in ⌀10 cm dishes (1 dish per condition) in media supplemented with 450 nM doxycycline for 48 h to induce expression of Cas9. Cells were then transfected with 10 μg of plasmids containing ERK0, or ERK20 reporters (Wang et al, 2020) for 24 h using jetOPTIMUS® reagent according to the manufacturer's instructions (Avantor, 101000051). Before collection, cells were treated with 20 μM proteasome inhibitor MG132 or DMSO as a control for 3 h. For collection, cells were washed in cold PBS, lysed on the plate with ice-cold lysis buffer (20 mM HEPES pH 7.3, 120 mM KCl, 5 mM MgCl$_2$, 1% NP-40, 1× EDTA-free protease inhibitor cocktail) supplemented with 10 mM N-ethylmaleimide (NEM). Cell lysates were further incubated on ice for 10 min with intermittent vortexing, passed through a 27 G needle three times, and clarified in a tabletop centrifuge at maximum speed (20,000× g) for 20 min at 4 °C. TUBE was inducibly expressed in *E. coli* Rosetta cells from 6His-TEVb-HaloTR-TUBE plasmid in pET28a backbone (MRC PPU Dundee, obtained from MRC PPU reagents, https://mrcppureagents.dundee.ac.uk/product/37522, DU23799) at 16 °C overnight, bacterial pellet was lysed in lysis buffer (25 mM HEPES, pH 7.4, 150 mM NaCl, 0.5% NP-40 1× EDTA-free protease inhibitor cocktail) supplemented with benzonase, sonicated and cleared by 20,000× g centrifugation for 30 min at 4 °C. The cleared bacterial lysate was conjugated to Halo beads (Promega, G7281) overnight at 4 °C on an end-over-end rotator, washed five times with wash buffer (25 mM HEPES, pH 7.4, 1 M NaCl, 0.5% NP-40 1× EDTA-free protease inhibitor cocktail) and once with cell lysis buffer (20 mM HEPES pH 7.3, 120 mM KCl, 5 mM MgCl$_2$, 1% NP-

40, 1× EDTA-free protease inhibitor cocktail, 10 mM NEM). Cleared cell lysates were incubated with conjugated Halo-TUBE beads for 3 h at 4 °C on an end-over-end rotator, washed three times with wash buffer (20 mM HEPES pH 7.3, 495 mM KCl, 5 mM MgCl$_2$, 1% NP-40, 1× EDTA-free protease inhibitor cocktail) supplemented with 10 mM N-ethylmaleimide (NEM) and eluted at 95 °C for 5 min in 1× SDS sample buffer with DTT added to a final concentration of 10 mM.

### Plant experiments

All *Arabidopsis thaliana* lines used in this study originate from the Columbia (Col-0) ecotype. Mutant lines used in this study are listed in Dataset EV2. Seedlings were grown in liquid 1/2 MS medium containing 1% sucrose under 16 h light/8 h dark photoperiod for 7 days with shaking at 80 rpm. Seven-day-old seedling grown in liquid 1/2 MS medium were treated with 100 μM anisomycin (ANS) for 16 h under continuous light with shaking at 80 rpm. An equal volume of pure dimethyl sulfoxide (DMSO) was added as control. Subsequently, seedlings were then frozen in liquid nitrogen after chemical treatment and homogenized for western blotting. The total protein was extracted with Grinding Buffer (50 mM Tris-HCl, 150 mM NaCl, 1% glycerol, 0.5% NP-40, 1.5 mM MgCl$_2$, 1× protease inhibitor cocktail). The SDS loading buffer was added to lysates, and the samples then were boiled at 95 °C for 10 min. 10 μg of sample was loaded per lane. SDS-PAGE was performed using gradient 4–20% Mini-PROTEAN TGX Precast Protein Gels (BioRad). Blotting on nitrocellulose membranes was performed using a semi-dry Turbo Transfer Blot System (BioRad). The membranes were blocked with 5% skimmed milk in TBS and 0.1% Tween 20 (TBS-T) for 1 h at room temperature. Subsequently, the membranes were incubated with primary antibody, followed by incubation with secondary antibody conjugated to horseradish peroxidase (HRP). After three times 10 min washes with TBS-T, the immune-reaction was developed using ECL SuperSignal West Femto (Thermo) and detected with iBright Imaging System (Thermo). Protein bands intensity was quantified using the iBright Imaging analysis System (Thermo). The average relative intensities and a standard error were calculated from three biological replicates.

### Graphics

Figs. 1A, 4C, 6 and synopsis were created with BioRender.com.

### Statistical analysis and reproducibility

All experiments were performed at least three times unless otherwise indicated in figure legends. Statistical analysis was performed in GraphPad Prism 10 software using unpaired two-sided Student *t*-test.

## Data availability

Mass spectrometry proteomics data have been deposited at the ProteomeXchange Consortium via the PRIDE partner repository (Perez-Riverol et al, 2022) with the dataset identifier PXD071028.

The source data of this paper are collected in the following database record: biostudies:S-SCDT-10_1038-S44318-026-00753-9.

## Peer review information

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

## Acknowledgements

We thank Kitti Csalyi and Thomas Sauer at the Max Perutz Labs Biooptics FACS facility, and Elisabeth Roitinger at the Vienna BioCenter Proteomics Core Facility, for their help. We thank Gijs Versteeg (Max Perutz Labs, Vienna) and Johannes Zuber (IMP, Vienna) for assistance with the iCas9 RKO cell lines and lentiviral transduction, and Yihong Ye (NIH, Bethesda) for the CytoK0, CytoK20, ERK0, and ERK20 plasmids. This research was funded in whole or in part by the Austrian Science Fund (FWF) [FWF-W1261, FWF-DOC 177B] to GEK, the [FWF-SFB F79] and Vienna Science and Technology Fund, WWTF-LS21 to GEK and YD, and European Research Council Grant (Project number: 101043370) to YD. ASA and MM are supported by the DOC fellowship of the Austrian Academy of Sciences. For open access purposes, the author has applied a CC BY public copyright license to any author-accepted manuscript version arising from this submission.

## Author contributions

**Milica Mihailovic**: Conceptualization; Formal analysis; Validation; Investigation; Visualization; Methodology; Writing—original draft; Writing—review and editing. **Aleksandra S Anisimova**: Conceptualization; Formal analysis; Validation; Investigation; Visualization; Methodology; Writing—review and editing. **Bu Erte**: Conceptualization; Formal analysis; Validation; Investigation; Visualization; Methodology; Writing—review and editing. **Ni Zhan**: Conceptualization; Formal analysis; Validation; Investigation; Visualization; Methodology; Writing—review and editing. **Ioanna Styliara**: Formal analysis; Investigation; Methodology. **Yasin Dagdas**: Conceptualization; Supervision; Funding acquisition; Investigation; Methodology; Project administration; Writing—review and editing. **Gülsün Elif Karagöz**: Conceptualization; Supervision; Funding acquisition; Investigation; Writing—original draft; Project administration; Writing—review and editing.

Source data underlying figure panels in this paper may have individual authorship assigned. Where available, figure panel/source data authorship is listed in the following database record: biostudies:S-SCDT-10_1038-S44318-026-00753-9.

## Disclosure and competing interests statement

The authors declare no competing interests.

# Expanded View Figures

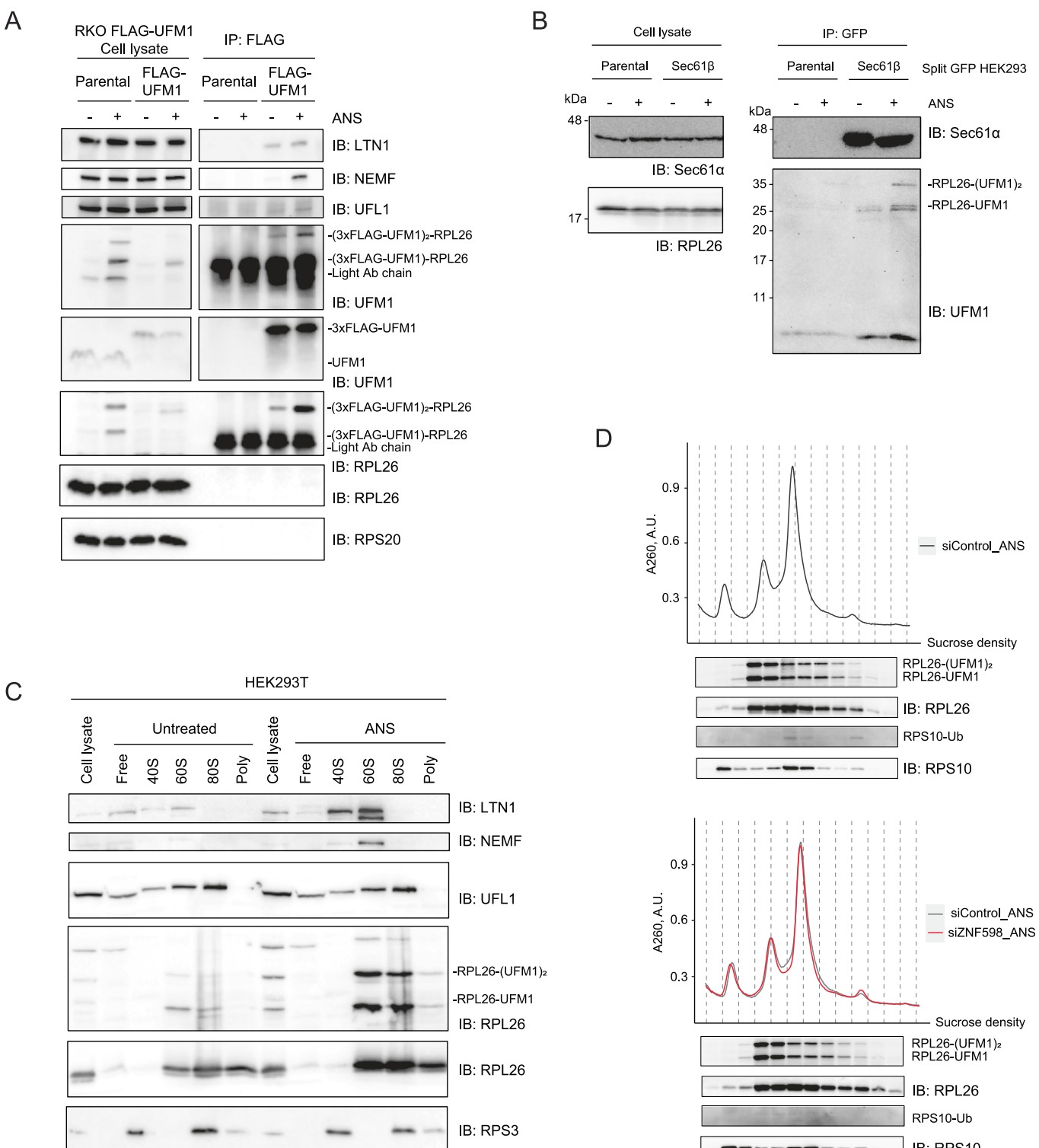

**Figure EV1. The RQC and UFM1 E3 ligase machinery associate with stalled ribosomes.**

(A) Immunoblot analysis of FLAG IP eluates upon 200 nM 1 h ANS treatment or from untreated RKO parental or RKO FLAG-UFM1 cells. (B) Immunoblot analysis of GFP IP eluates upon 4 μM 3 h ANS treatment or from untreated HEK293 split GFP parental or HEK293 Sec61β-split GFP cell lines. (C) Immunoblots of selected sucrose fractions of HEK293T cell lysates upon 200 nM 1 h ANS treatment or from untreated control condition. (D) Polysome profiles from HCT116 cells upon 72 h knockdown of control siRNA (top panel) or ZNF598 (bottom panel), accompanied by immunoblots showing RPL26 and RPS10 protein levels from corresponding fractions.

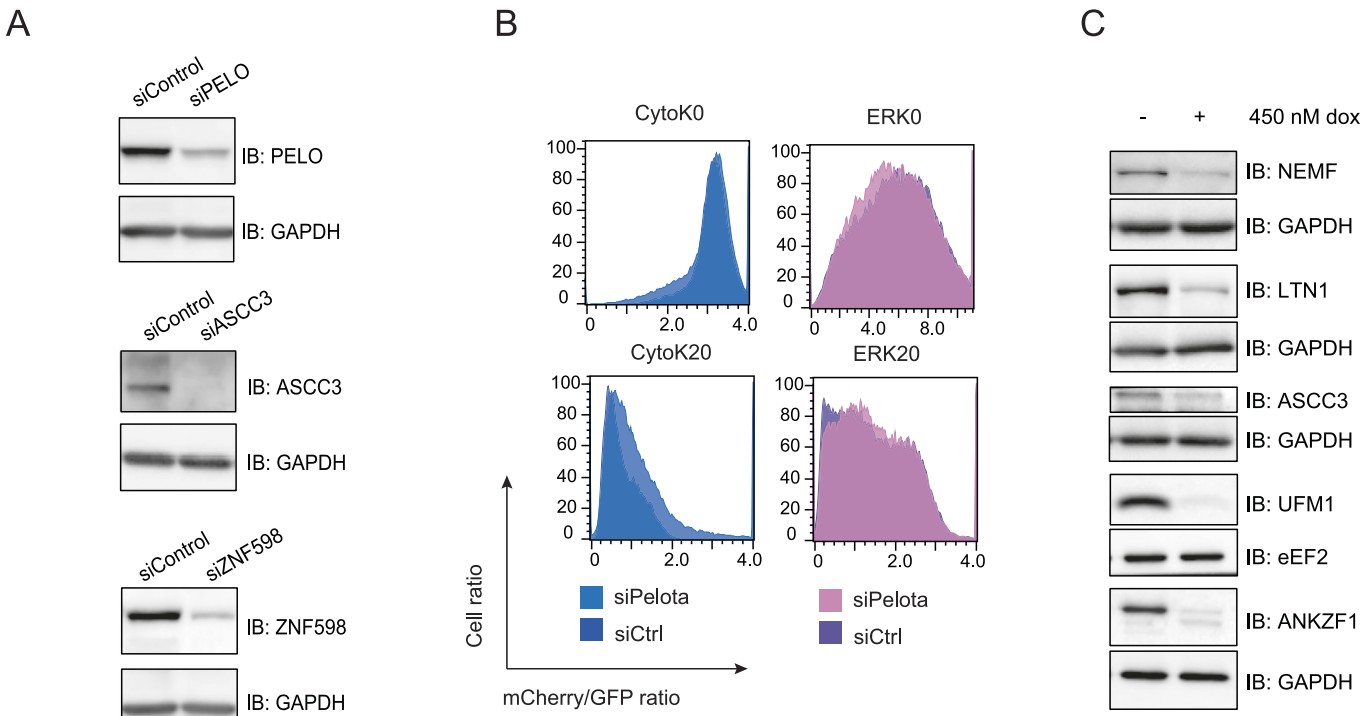

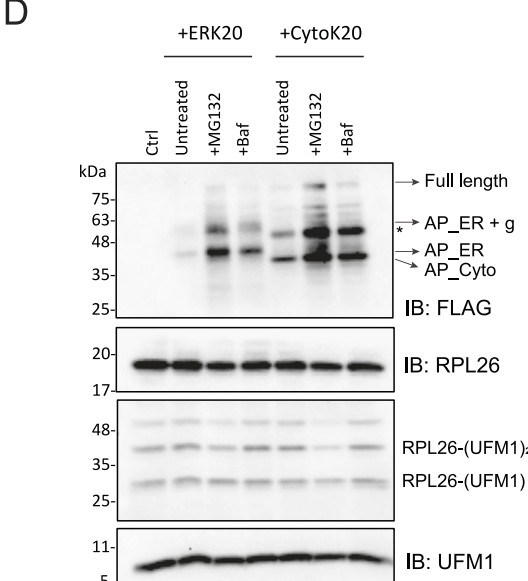

**Figure EV2.   The ER stalling reporter is primarily stabilized by a proteasomal inhibitor.**

(A) Immunoblots showing knockdowns of RQC proteins Pelota, ASCC3, or ZNF598 upon 72 h siRNA treatment in HCT116 cells. GAPDH is used as a loading control. (B) Readthrough of reporters from Fig. 2A shown by mCherry/GFP ratio measured by FACS after 24 h expression in HCT116 cells upon siRNA-mediated knockdown of Pelota compared to non-targeting control siRNA. (C) Immunoblots showing knockouts of RQC proteins NEMF, LTN1, ASCC3, ANKZF1, or UFM1 knockout upon 48 h dox treatment in RKO cells. GAPDH or eEF2 are used as a loading control. (D) Immunoblots showing reporter accumulation upon 24 h expression of CytoK20 or ERK20 in HCT116 cells after treatment with either 20 nM MG132 for 3 h, 10 nM Baf for 16 h, or untreated as a control. AP arrested peptide, * degradation products.

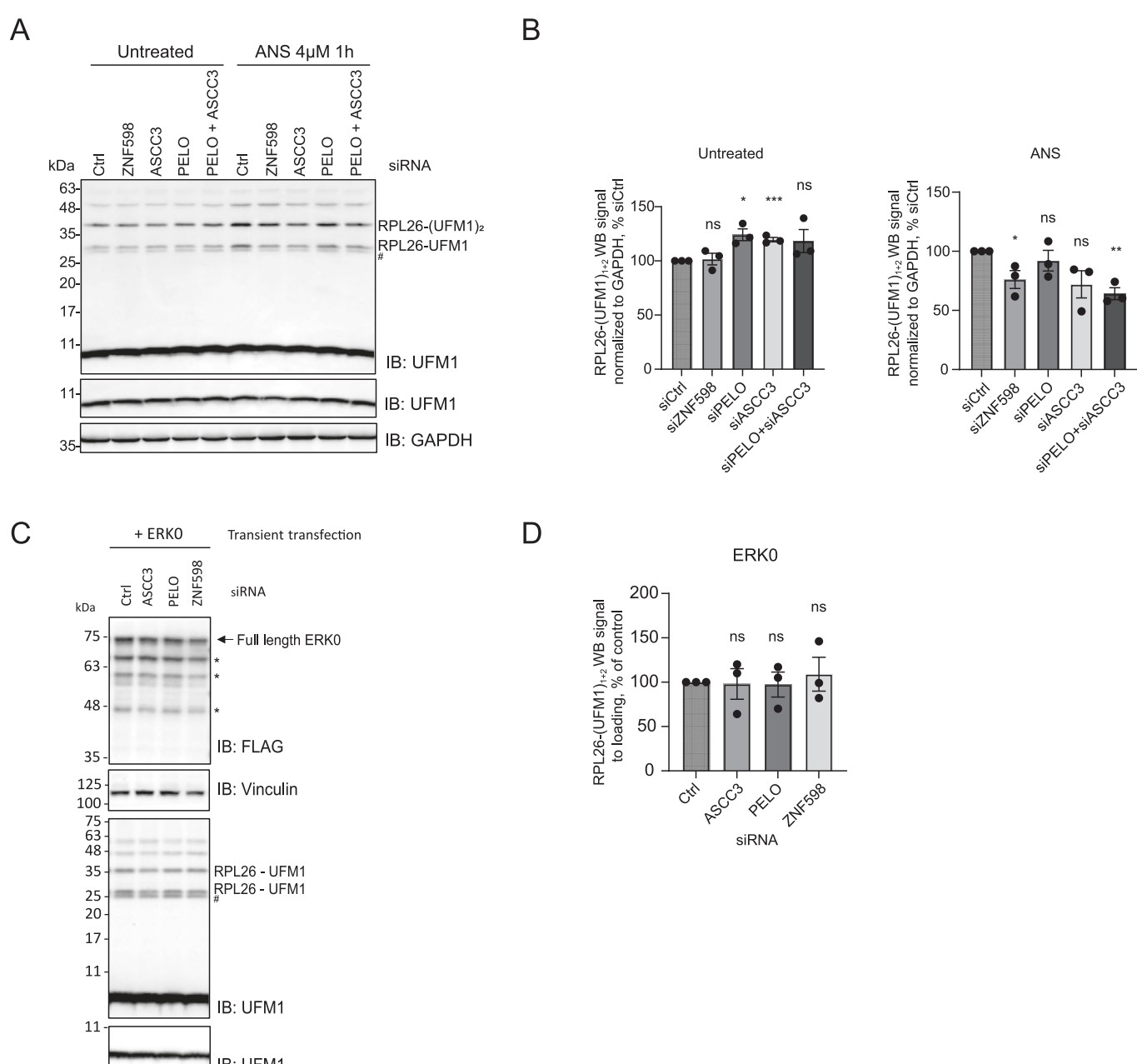

**Figure EV3. Ribosomal splitting precedes UFMylation of RPL26.**

(A) UFMylation levels visualized by UFM1 immunoblot in untreated or 1 h 4 μM ANS treated HCT116 cells upon 72 h siRNA-mediated knockdown of RQC components or non-targeting control. (B) Quantification of (A) with unpaired two-sided Student *t*-test, *n* = 3. Error bars represent SEM. (C) UFMylation levels visualized by UFM1 immunoblot in HCT116 cells upon 72 h siRNA-mediated knockdown of RQC components or non-targeting control, and 24 h expression of ERK0. (D) Quantification of (C) with unpaired two-sided Student *t*-test, *n* = 3. Error bars represent SEM. ****$P \leq 0.0001$, ***$P \leq 0.001$, **$P \leq 0.01$, *$P \leq 0.05$. List of complete *P* values available in Dataset EV2. # - UFC1-UFM1 complex, * degradation products.

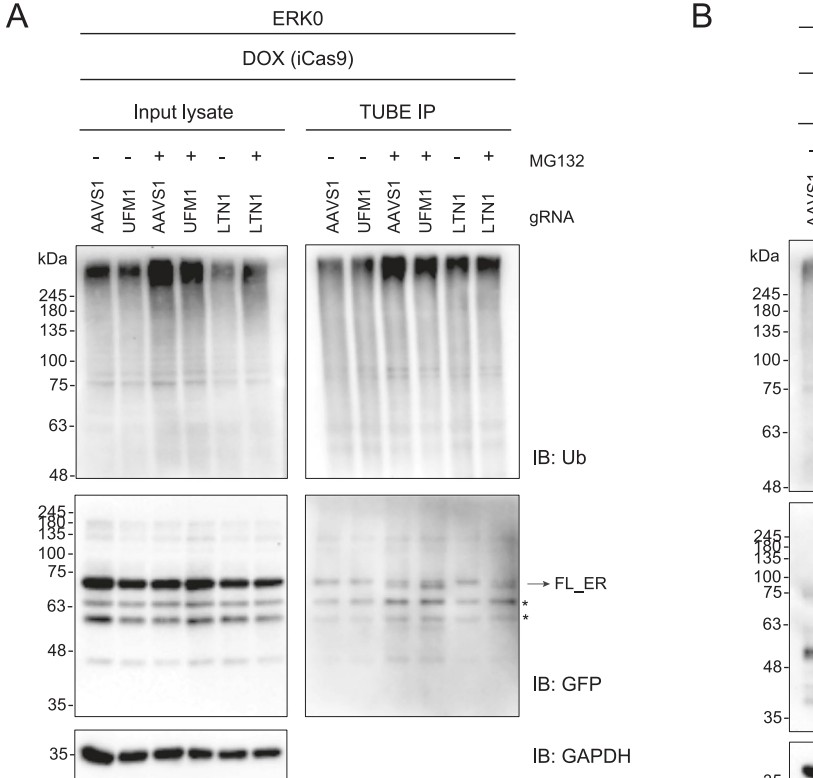

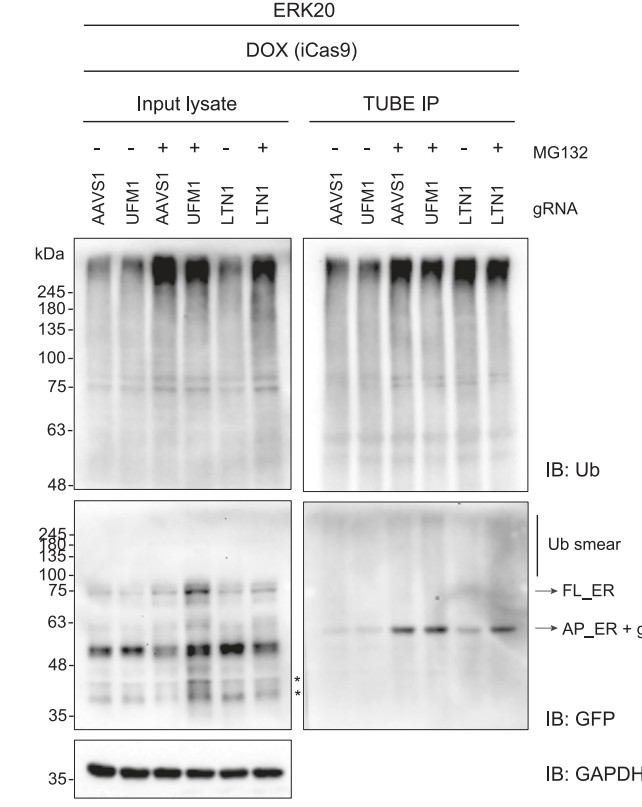

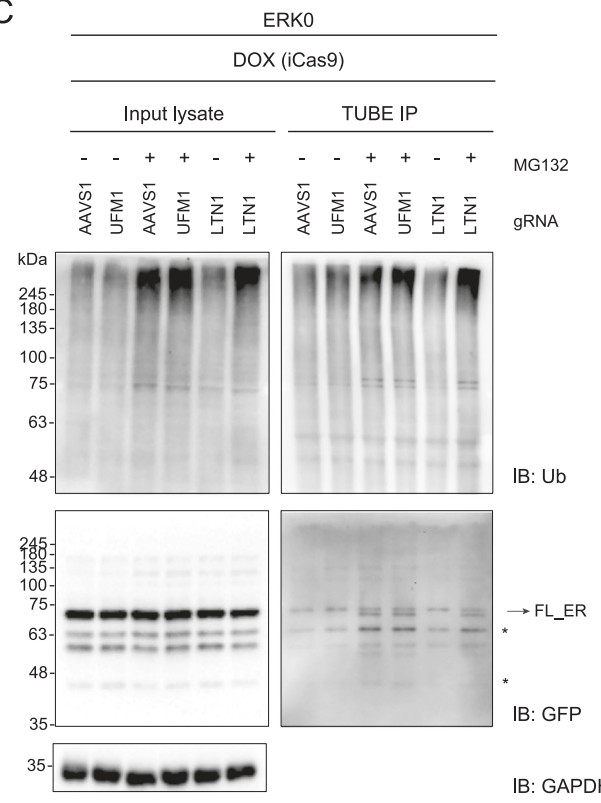

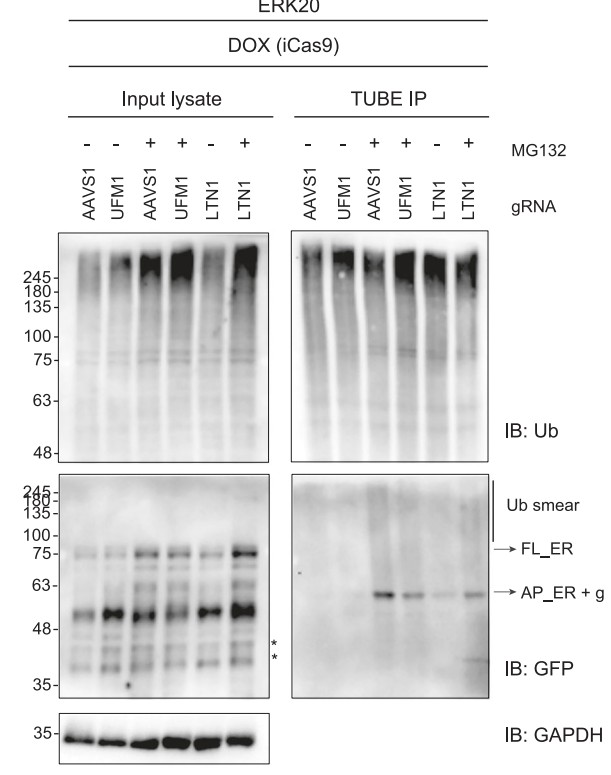

**Figure EV4. The loss of LTN1 or UFM1 does not impair the ubiquitination of ERK2O.**

TUBE assays were performed from RKO iCas9 AAVS1 (non-targeting control), LTN1 or UFM1 KO cell line upon 24 h expression of ERK0 (**A**, **C**) or ERK2O reporter (**B**, **D**) and 3 h treatment with 20 µM proteasome inhibitor MG132 or DMSO as control. Protein levels are analyzed by immunoblotting. Representative duplicates. FL full length, AP arrested peptide, g glycosylation, * degradation products.

