## [Peer Review File · The EMBO Journal]

The coordinated action of UFMylation and the RQC pathways clears arrested polypeptides at the ER

Milica Mihailovic, Aleksandra Anisimova, Bu Erte, Ni Zhan, Ioanna Styliara, Yasin Dagdas, and Gülsün Karagöz

Corresponding author(s): Gülsün Karagöz (guelsuen.karagoez@meduniwien.ac.at) , Yasin Dagdas (yasin.dagdas@cos.uni-heidelberg.de)

Review Timeline:

Submission Date:	15th May 25
Editorial Decision:	29th Jul 25
Revision Received:	6th Dec 25
Editorial Decision:	30th Jan 26
Revision Received:	6th Feb 26
Accepted:	25th Feb 26

Editor: *Cornelius Schneider*

Transaction Report:

Dear Dr. Karagöz,

Thank you for submitting your manuscript for consideration by the EMBO Journal. It has now been seen by three referees whose comments are shown below.

Thank you also for also our productive discussions and for agreeing to perform experiments to provide additional insight. I would therefore like to invite you to submit a revised version of the manuscript, addressing the comments of all three reviewers as discussed in our meeting. I should add that it is EMBO Journal policy to allow only a single round of revision, and acceptance of your manuscript will therefore depend on the completeness of your responses in this revised version.

Thank you for the opportunity to consider your work for publication. I look forward to your revision.

Yours sincerely,

Cornelius Schneider, PhD
Editor
The EMBO Journal
c.schneider@embojournal.org

Please remember: Digital image enhancement is acceptable practice, as long as it accurately represents the original data and conforms to community standards. If a figure has been subjected to significant electronic manipulation, this must be noted in the figure legend or in the 'Materials and Methods' section. The editors reserve the right to request original versions of figures and

the original images that were used to assemble the figure.

We realize that it is difficult to revise to a specific deadline. In the interest of protecting the conceptual advance provided by the work, we recommend a revision within 3 months (27th Oct 2025). Please discuss the revision progress ahead of this time with the editor if you require more time to complete the revisions. Use the link below to submit your revision:

Referee #1:

UFM1 is a ubiquitin-like protein whose most prevalent and best-characterized substrate is the ribosomal protein RPL26. Recent studies have significantly advanced our understanding of the UFMylation pathway, including its molecular mechanisms, structural features, and physiological roles. The UFM1 E3 ligase complex is localized to the endoplasmic reticulum (ER) via one of its ER membrane-anchored subunits. There, it targets ribosomes stalled at the translocon for UFMylation. Consistent with knowledge that stalled ribosomes and their associated nascent chains are substrates for ribosome-associated quality control (RQC), several lines of evidence have linked UFMylation and RQC. However, the precise molecular relationship between these two pathways has remained elusive.

In 2024, two high-profile structural biology studies proposed that UFMylation acts by releasing 60S subunits obstructing the translocon after the dissociation of RQC components and nascent chains. However, this model does not reconcile with earlier observations that loss of UFMylation leads to an accumulation of RQC substrates.

The manuscript by Mihailovic and colleagues offers a mechanistically satisfactory alternative. They propose that UFMylation induces a conformational change in the translocon-associated 60S subunit that stabilizes the binding of the RQC component LTN1, thereby facilitating its access to the nascent chain. The data presented are of high technical quality and support this interpretation. I support publication in EMBO Journal, provided that the authors address the following minor comments:

1. The first half of the manuscript largely recapitulates findings previously reported by others. The authors should clearly indicate which results are confirmatory and cite relevant prior work accordingly. For example, the increase in RPL26 UFMylation upon LTN1 knockout was first reported, to my knowledge, by Tanaka's group in 2023 (PMID: 36917672), and should be cited.
2. The terms "early RQC" and "late RQC" are non-standard and potentially confusing. As RQC was originally defined as the LTN1-containing complex acting on 60S subunits, the authors should replace these terms with clearer language such as "ribosome rescue" and "RQC," respectively.
3. Lines 168-169: "These data also indicated that UFMylation does not impair and probably precedes NEMF binding." The rationale for the word "probably" is unclear and should be briefly explained.
4. Lines 234-235: "We anticipate that depletion of NEMF and LTN1 impairs the clearance of the stalled ribosomes at the ER." Given that complete blockade was not demonstrated, alternative words to "impairs" such as "decreases" or "interferes with" would be more accurate. In fact, it is possible that other mechanisms-such as peptidyl-tRNA cleavage or hydrolysis-might contribute to ribosome clearance. While not required for publication, assessing the effect of ANKZF1 depletion could offer new insights into their findings.
5. Line 343-344: The conclusion that "by destabilizing the translocon-60S association, UFMylation allows access of LTN1 to the nascent chain" is plausible and could be further supported by monitoring nascent chain ubiquitination on a stalling reporter. While this is not a requirement for publication, it would strengthen the mechanistic link.
6. Lines 343-344 (continued): The authors should note that their data also suggest that, by destabilizing the translocon-60S association, UFMylation may allow positioning of the LTN1 C-terminus (that includes the RING domain) in its canonical binding site in the vicinity of the exit tunnel, stabilizing its interaction with the complex.
7. A key observation from this and prior work is that RPL26-UFM1 levels increase upon LTN1 loss. What is the mechanism? Does LTN1 binding promote de-UFMylation? In the study by Tanaka's group, expression of an LTN1 mutant lacking the RING domain (LTN1 Δ RING) reduced RPL26 UFMylation in LTN1 knockout cells, suggesting that ubiquitination per se is not required for this effect. Although not required for acceptance, this would further strengthen the story.

Referee #2:

This study investigates the relationship between the UFMylation pathway and the RQC pathway for degradation of nascent polypeptides stalled during their translocation into the ER. The authors report that UFMylated ribosomes can co-associate with at least some of the RQC machinery, that the absence of UFMylation selectively stabilizes an ER RQC reporter but not a cytosolic RQC reporter, that UFMylation is reduced if ribosome collisions and splitting are prevented, and that UFMylation not appreciably affected (or can increase) by loss of later RQC factors. These observations are used to propose that UFMylation precedes 60S engagement of stalled ER RQC substrates by NEMF and LTN1. Together with earlier findings in the literature, a model is proposed where UFMylation acts on ER-associated 60S peptidyl-tRNA complexes that have been acted on by ZNF598 and ASC-1 complex.

The main contribution of this study beyond earlier work is to propose that UFMylation and RQC work together on the same 60S complexes to mediate nascent chain degradation, with UFMylation acting first to facilitate the action of RQC factors.

Rigorously ordering the sequence of events from ribosome stalling at the ER through to nascent chain degradation by the proteasome is an important goal. However, this study falls short of this aim in a number of ways, has a number of over-interpretations, and seems to have been hastily assembled (as evidenced by numerous errors, both minor and gross). The paper is in need of revisions that involves some experimental work, re-writing, and re-interpretation to be suited for publication.

Major points:

1) A major claim in the paper is that the UFMylation machinery and RQC machinery assemble on the same 60S-peptidyl-tRNA arising from stalled ER-translocating substrates. This is likely true, but is not well demonstrated in this study. Fig. 1 is intended to show that UFMylated ribosomes contain both the UFMylation machinery and RQC machinery. However, it is rather incomplete and poorly validated as follows:

1a) The UFMylated products pulled down via tagged UFM1 are never shown to be 60S complexes. This is an important omission given that UFMylated RPL26 can be detected in 60S, 80S, and larger fractions in Fig. S2C. I think it is important to document using better resolution gradients and blotting for both 60S and 40S ribosomal proteins that the products of the pulldown are primarily, if not exclusively, 60S complexes. If 80S products are detected, then some explanation and adjustment of the conclusions is warranted.

1b) The MS data shown in Fig. 1B does not indicate what is statistically significant, and as far as I can tell, does not have a negative control using cells with untagged UFM1. Such a control was evidently performed as indicated in the supplementary table, although evidently only once and only under ANS conditions. This makes for a rather poorly controlled analysis, and a reader cannot appreciate what is specific and what is background, and cannot deduce what is a reliable interaction. Of course, it would be ideal to repeat this experiment with suitable negative controls for both the untreated and ANS conditions, but if the authors don't want to do this, they should present the data in a more sensible manner. I would recommend that the authors present a standard volcano plot with statistical values for the ANS condition comparing tagged versus untagged pulldowns. On this plot, they should colour all the 40S and 60S ribosomal proteins in separate colours, all the translation-related factors in another colour, and label the UFMylation and RQC factors. This way, a reader can appreciate (and judge) the quality of the pulldown and can also judge whether a UFM1 pulldown specifically pulls out 60S over 40S, and also see which UFMylation and RQC factors are co-purified. In a separate panel, the authors can then present the comparison of the untreated and ANS samples. Right now, I have a very hard time evaluating the experiment at all.

1c) A pulldown experiment as in Fig. 1B should be blotted for all the UFMylation and RQC factors (along with 60S and 40S markers) and Sec61 to validate the MS analysis.

1d) There are a number of aspects of Fig. 1C that are confusing to me. According to the data shown in panel B, the cushion-normalised ANS/Unt ratio should be 1 if the point falls on the diagonal, would be higher than 1 if it is above the diagonal, and lower than 1 if it is below the diagonal. Yet, DDRGK1, which is clearly below the diagonal in panel B, is shown in panel C to have a ANS/Unt ratio of 4 in both replicates. Similarly, CDK5RAP3 is only a little above the diagonal and nowhere close to NEMF or UFM1 in panel B, yet in panel C, is shown to be roughly the same in both replicates. Thus, the data shown in B and that depicted in C do not seem to match, even though the legend indicates that panel C is a bar graph using data from panel B plus another replicate. Looking at the supplementary table also seems to indicate that what is plotted in panel C is not normalised to the cushion samples. Furthermore, it seems that the same untreated sample is used as the denominator for the two replicates, indicating that what is plotted in panel C is not actually two independent replicates. All of these things need to be clarified and the analysis should be more transparent and rigorous.

2) The data in Fig. 2 basically reproduces earlier experiments, and it is important that the authors acknowledge this. In particular, the fact that UFMylation is needed for effective degradation of an ER-RQC substrate but not a cytosolic RQC substrate was nicely documented in PMID 37036982 and 31595041. Furthermore, the heading "RQC factors and UFMylation machinery collaborate..." is not supported by what is shown in Fig. 2. The data only allow one to conclude that both sets of factors are needed for maximal degradation, but there is nothing to indicate that they collaborate. Please adjust the language here.

3) There are a number of places where the authors blur the lines between what their data show versus what they infer from earlier work to interpret their data. For example, starting on line 270, they say they show that stalled ribosomes are recognised by ZNF598 and split by the ASC-1 complex, but in fact, they never show this. Instead, this is inferred from earlier studies that investigated this, and this information is then used to conclude that Ufmylation depends (partially) on splitting. What is actually shown by the authors is that Ufmylation is diminished in the absence of either ZNF598 or ASC-1 complex. The implication of this observation is that recognition of collisions (mediated by ZNF598 - PMID) and subsequent ribosome splitting (mediated by ASC-1 complex - PMID) are likely to be prerequisites for Ufmylation. I strongly encourage the authors to be far more precise in their language and interpretations.

4) In Fig. 3E, it is stated the ERK20 increases Ufmylation, whereas CytoK20 does not. But in this experiment, there is no sample for cells not expressing a stalling reporter. So it is unclear to me what is being taken as baseline here. Perhaps the authors mean to say that ERK20 shows higher Ufmylation compared to CytoK20? Please clarify.

5) The authors often make definitive statements when they are not warranted. For example, they say splitting is required for Ufmylation, but there is no solid evidence to support this claim. Instead, they see a PARTIAL reduction in Ufmylation when the ASC-1 complex or ZNF598 are absent. They need to acknowledge this partial effect, and ideally, provide some explanation for why the effects are partial. Again, as in the previous comment, more care is needed in how the data are interpreted and how the paper is written.

6) The effect of Baf is more substantial for the ER reporter than on the cytosolic one. Perhaps mention this observation, and the possibility that the ER reporter can be potentially released into the secretory pathway and trafficked to lysosomes for degradation. Also, Baf is not really an autophagy inhibitor; it is better to say it is an inhibitor of lysosomal degradation.

7) Pelota knockdown sometimes decreases Ufmylation, but other times has no effect. Can the authors please clarify the reason(s) for this.

8) The substrate pulldown experiments in Fig. 4 should each be analysed more thoroughly by blotting the samples for other relevant factors in addition to what is shown: 40S protein, NEMF, and LTN1. Furthermore, there is no control sample lacking a GFP-tagged substrate, so one cannot judge what background recovery would look like. The experiment as presented is rather incomplete and does not allow for unambiguous conclusions.

9) The legends of Fig. S1 don't match what is shown in the figure. The legend has panels A-E, the figure only panels A-D; also some of the panels don't match the legend. Please fix.

Referee #3:

This manuscript demonstrates the interplay between the ribosome-associated quality control (RQC) machinery and UFMylation in the clearance of arrested peptides at the endoplasmic reticulum (ER). The key findings are that dissociation of the 80S ribosome into the 60S subunit is a prerequisite for UFMylation of RPL26, a component of the 60S subunit, and that the UFM1 E3 ligase complex subsequently facilitates the recruitment of late RQC factors NEMF and LTN1. The experiments are well-designed, and the results are clearly presented. However, the majority of these core findings have already been reported (PMID: 38383785, PMID: 38383789, PMID: 40315331), and the novelty here is largely limited to implicating early RQC factors in the process. Moreover, the involvement of these early factors has already been anticipated based on prior studies. While I appreciate that this study strengthens prior structural observations with additional genetic and biochemical evidence, the overall impact of the findings is limited. In its current form, I think the manuscript does not meet the novelty and significance criteria for The EMBO Journal.

Minor Comments

1. The authors should indicate RPL26 and LTN1 in Figure 1B.

2. The authors should cite the work from the Beckmann lab (PMID: 40315331), which demonstrated that the UFM1 E3 ligase complex forms a complex with the late RQC factors NEMF and LTN1.

We want to thank the referees for their time and invaluable input. We appreciate their thorough assessment and their constructive criticism. We conducted most of the experiments based on their feedback and edited the manuscript accordingly. We believe that addressing their points improved our manuscript. Please see below our point-by-point response to the referees' comments and revised manuscript, where all the edited text is marked in blue.

Referee #1:

UFM1 is a ubiquitin-like protein whose most prevalent and best-characterized substrate is the ribosomal protein RPL26. Recent studies have significantly advanced our understanding of the UFMylation pathway, including its molecular mechanisms, structural features, and physiological roles. The UFM1 E3 ligase complex is localized to the endoplasmic reticulum (ER) via one of its ER membrane-anchored subunits. There, it targets ribosomes stalled at the translocon for UFMylation. Consistent with knowledge that stalled ribosomes and their associated nascent chains are substrates for ribosome-associated quality control (RQC), several lines of evidence have linked UFMylation and RQC. However, the precise molecular relationship between these two pathways has remained elusive.

In 2024, two high-profile structural biology studies proposed that UFMylation acts by releasing 60S subunits that obstruct the translocon after dissociation of RQC components and nascent chains. However, this model does not reconcile with earlier observations that loss of UFMylation leads to an accumulation of RQC substrates.

The manuscript by Mihailovic and colleagues offers a mechanistically satisfactory alternative. They propose that UFMylation induces a conformational change in the translocon-associated 60S subunit, stabilizing the binding of the RQC component LTN1 and thereby facilitating its access to the nascent chain. The data presented are of high technical quality and support this interpretation. I support publication in EMBO Journal, provided that the authors address the following minor comments:

We thank Referee #1 for the constructive and detailed feedback. Please see our response to individual points.

1. The first half of the manuscript largely recapitulates findings previously reported by others. The authors should clearly indicate which results are confirmatory and cite relevant prior work accordingly. For example, the increase in RPL26 UFMylation upon LTN1 knockout was first reported, to my knowledge, by Tanaka's group in 2023 (PMID: 36917672), and should be cited.

We want to thank Referee #1 for bringing this paper to our attention. We have now gone through the manuscript and added the relevant references, including the manuscript PMID: 36917672, for which citations were missing or incomplete.

2. The terms "early RQC" and "late RQC" are non-standard and potentially confusing. As RQC was originally defined as the LTN1-containing complex acting on 60S subunits, the authors should replace these terms with clearer language such as "ribosome rescue" and "RQC," respectively.

Thanks for bringing this to our attention. We agree that this can be misleading. We have now defined these stages in the introduction and changed the terminology throughout the paper.

3. Lines 168-169: "These data also indicated that UFMylation does not impair and probably precedes NEMF binding." The rationale for the word "probably" is unclear and should be briefly explained.

At that point in the manuscript, we did not present the data on dependency. We have now removed this phrase to avoid confusion. Later in the manuscript, we explain this claim in lines 298-301 and lines 381-388, supported by experimental data shown in Figures 3 and 4.

4. Lines 234-235: "We anticipate that depletion of NEMF and LTN1 impairs the clearance of the stalled ribosomes at the ER." Given that complete blockade was not demonstrated, alternative words to "impairs" such as "decreases" or "interferes with" would be more accurate. In fact, it is possible that other mechanisms-such as peptidyl-tRNA cleavage or hydrolysis-might contribute to ribosome clearance. While not required for publication, assessing the effect of ANKZF1 depletion could offer new insights into their findings.

We assessed the possible role of ANKZF1 in the degradation of ER-stalled peptides in Figure 2D, E, showing that ANKZF1 depletion impacts the stability of the ERK20 construct, suggesting that peptidyl-tRNA cleavage or hydrolysis contributes to the clearance of arrested polypeptides in the ER. We now also examined the impact of ANKZF1 depletion on UFMylation of RPL26 and did not observe a strong effect Figure 3J-K. Together with our data on the effect of ANKZF1 depletion on stalled polypeptides, these findings suggest that ANKZF1 might act on stalled ribosomes after the deUFMylation of RPL26 to cleave the peptidyl-tRNA bond, which is the last step in the clearance of the arrested polypeptides. Please also see lines 237-238 and 301-304.

5. Line 343-344: The conclusion that "by destabilizing the translocon-60S association, UFMylation allows access of LTN1 to the nascent chain" is plausible and could be further supported by monitoring nascent chain ubiquitination on a stalling reporter. While this is not a requirement for publication, it would strengthen the mechanistic link.

Thank you very much for the suggestion. To address this point, we performed Tandem Ubiquitin Binding Entity (TUBE) pulldowns followed by WB analyses to determine whether ubiquitination of the ER-stalling reporter ERK20 or the control ERK0 is affected by UFM1 or LTN1 depletion. We do not observe impairment of ERK20 ubiquitination upon either UFM1 or LTN1 depletion. However, we find it very challenging to quantify small changes with these assays. We repeated this experiment 7 times for UFM1 depletion and observed that UFM1 depletion decreased

ERK20 ubiquitination in 2 experiments (50% and 75% of control levels), but in the other five experiments, UFM1 depletion had no observable effect. Therefore, we can conclude that depletion of either UFM1 or LTN1 does not impact ERK20 ubiquitination. We anticipate that when the primary clearance pathways are depleted, other E3s may begin ubiquitinating stalled polypeptides. Please see the representative triplicate data and quantifications in Figure 5A, B, and Figure EV5A-D, lines 402-419.

6. Lines 343-344 (continued): The authors should note that their data also suggest that, by destabilizing the translocon-60S association, UFMylation may allow positioning of the LTN1 C-terminus (that includes the RING domain) in its canonical binding site in the vicinity of the exit tunnel, stabilizing its interaction with the complex.

We thank the reviewer for the suggestion and agree that this clearly describes the mechanistic model. We have added this sentence to the revised version. Please see lines 397-400.

7. A key observation from this and prior work is that RPL26-UFM1 levels increase upon LTN1 loss. What is the mechanism? Does LTN1 binding promote de-UFMylation? In the study by Tanaka's group, expression of an LTN1 mutant lacking the RING domain (LTN1 Δ RING) reduced RPL26 UFMylation in LTN1 knockout cells, suggesting that ubiquitination per se is not required for this effect. Although not required for acceptance, this would further strengthen the story.

To address this point, we used lentiviral transduction to generate stable cell lines expressing GFP-control, wild-type LTN1, or the LTN1 Δ RING mutants in cells expressing the doxycycline-inducible iCas9 system to deplete LTN1. This strategy ensures short-term LTN1 depletion and avoids compensatory effects. In those experiments, upon depletion of LTN1 using doxycycline, we observe accumulation of UFMylated RPL26, and this phenotype is not observed in cells expressing either LTN1 or LTN1 Δ RING, recapitulating the data from Tanaka's lab (Response Figure 1A and 1 B). However, surprisingly, we see rescue of both ERK20 and CytoK20 upon expression of LTN1 Δ RING in those cells, contrasting earlier findings showing the importance of the RING domain for the degradation of the cytosolic stalling reporter, consistent with the mechanism of action of RING-type E3 ligases^{1,2} (Response Figure 1D-G). We confirmed that endogenous LTN1 depletion is equally efficient across all three cell lines, ruling out that any remaining endogenous LTN1 could rescue the phenotype in those cells (Response Figure 1C). We would prefer not to include this data in the manuscript until we better understand our system and further dissect this discrepancy.

We also attempted to assess the deUFMylation efficiency in the presence and absence of LTN1 using emetine/cycloheximide washout experiments. Unfortunately, we did not achieve effective RPL26-deUFMylation even after 3 hours of washout, preventing us from directly assessing deUFMylation kinetics in this system in suitable timelines.

To further dissect the mechanistic cross-talk between LTN1 binding with RPL26 deUFMylation, we assessed the impact of loss of UFSP2 on LTN1 association with the 60S-peptidyl-tRNA complexes in GFP-nascent chain pulldowns after sucrose cushion. We show that, while short depletion of UFSP2 accumulates UFMylated RPL26, under those conditions the LTN1 association to stalled ribosomes is not affected, suggesting that deUFMylation neither precedes nor is required for LTN1 association (please see Figure 5C-E and lines 423-33).

Response Figure 1. LTN1 rescue effect on UFMylation and accumulation of ribosome stalling reporters. **A.** RPL26 UFMylation levels visualized by UFM1 immunoblot of cell lysates of RKO iCas9 LTN1 KO, RKO iCas9 LTN1 KO expressing LTN1 WT or ΔRING mutant upon 72h of doxycycline-induced Cas9 expression. **B.** Quantification of **A**, n=7. **C.** The efficiency of doxycycline-induced knockout of endogenous LTN1 in cell lines from **A**. visualized by qPCR analysis of endogenous LTN1 3'UTR (upper panel) or 5'UTR (lower panel), n=4. **D** and **E.** Stalling reporter accumulation levels visualized by GFP immunoblot of cell lysates of RKO iCas9 LTN1 KO, RKO iCas9 LTN1 KO

expressing LTN1 WT or Δ RING mutant upon 72h of doxycycline-induced Cas9 expression, and 24h of ERK0 and ERK20 (D) or CytoK20 (E) reporter transient transfection. F and G. Quantification of D and E, respectively. ERK0 n=3, ERK20 n=6 and CytoK20 n=4.

Referee #2:

This study investigates the relationship between the UFMylation pathway and the RQC pathway for degradation of nascent polypeptides stalled during their translocation into the ER. The authors report that UFMylated ribosomes can co-associate with at least some of the RQC machinery, that the absence of UFMylation selectively stabilizes an ER RQC reporter but not a cytosolic RQC reporter, that UFMylation is reduced if ribosome collisions and splitting are prevented, and that UFMylation not appreciably affected (or can increase) by loss of later RQC factors. These observations are used to propose that UFMylation precedes 60S engagement of stalled ER RQC substrates by NEMF and LTN1. Together with earlier findings in the literature, a model is proposed where UFMylation acts on ER-associated 60S peptidyl-tRNA complexes that have been acted on by ZNF598 and ASC-1 complex.

The main contribution of this study beyond earlier work is to propose that UFMylation and RQC work together on the same 60S complexes to mediate nascent chain degradation, with UFMylation acting first to facilitate the action of RQC factors.

Rigorously ordering the sequence of events from ribosome stalling at the ER through to nascent chain degradation by the proteasome is an important goal. However, this study fall short of this aim in a number of ways, has a number of over-interpretations, and seems to have been hastily assembled (as evidenced by numerous errors, both minor and gross). The paper is in need of revisions that involves some experimental work, re-writing, and re-interpretation to be suited for publication.

We thank Referee #2 for their time and their detailed assessment and critical feedback. Please see our response to their points below.

Major points:

1) A major claim in the paper is that the UFMylation machinery and RQC machinery assemble on the same 60S-peptidyl-tRNA arising from stalled ER-translocating substrates. This is likely true, but is not well demonstrated in this study. Fig. 1 is intended to show that UFMylated ribosomes contain both the UFMylation machinery and RQC machinery. However, it is rather incomplete and poorly validated as follows:

1a) The UFMylated products pulled down via tagged UFM1 are never shown to be 60S complexes. This is an important omission given that UFMylated RPL26 can be detected in 60S, 80S, and larger fractions in Fig. S2C. I think it is important to document using better resolution

gradients and blotting for both 60S and 40S ribosomal proteins that the products of the pulldown are primarily, if not exclusively, 60S complexes. If 80S products are detected, then some explanation and adjustment of the conclusions is warranted.

We thank the referee for the suggestion. To address this point, we now performed high-resolution sucrose gradients to assess co-fractionation of UFL1, NEMF, and LTN1 with ribosomes and ribosomal subunits. The UFL1 complex co-fractionates with 40S, 60S, and 80S, while NEMF and LTN1 co-migrate only with 60S, in line with earlier observations^{3,4} (Figure 1F, G, Figure EV1A). It is plausible that the UFL1 ligase complex also associates with the 40S subunit, as earlier data suggest that UFMylation occurs on small ribosomal proteins and translation initiation factors^{5,6}. However, as comigration does not prove direct association, further experiments are required to validate this. To address this during revision, we attempted to IP FLAG-UFM1-associated proteins from various fractions in sucrose gradients. To upscale the lysates, we used the SW-32 rotor instead of the SW-40, which allows loading larger volumes. However, we could not detect any interaction partner when we performed IP-WB analyses of distinct fractions from the gradients; we suspect this is due to dilution during the gradient runs.

1b) The MS data shown in Fig. 1B does not indicate what is statistically significant, and as far as I can tell, does not have a negative control using cells with untagged UFM1. Such a control was evidently performed as indicated in the supplementary table, although evidently only once and only under ANS conditions. This makes for a rather poorly controlled analysis, and a reader cannot appreciate what is specific and what is background, and cannot deduce what is a reliable interaction. Of course, it would be ideal to repeat this experiment with suitable negative controls for both the untreated and ANS conditions, but if the authors don't want to do this, they should present the data in a more sensible manner. I would recommend that the authors present a standard volcano plot with statistical values for the ANS condition comparing tagged versus untagged pulldowns. On this plot, they should colour all the 40S and 60S ribosomal proteins in separate colours, all the translation-related factors in another colour, and label the UFMylation and RQC factors. This way, a reader can appreciate (and judge) the quality of the pulldown and can also judge whether a UFM1 pulldown specifically pulls out 60S over 40S, and also see which UFMylation and RQC factors are co-purified. In a separate panel, the authors can then present the comparison of the untreated and ANS samples. Right now, I have a very hard time evaluating the experiment at all.

We agree with the referee that performing the pulldown experiments in triplicate to achieve statistical significance is crucial. We now performed triplicate pulldown experiments with IgG controls and presented the data as volcano plots, plotting all the components (Figure 1B, C). We observe a clear, statistically significant enrichment of UFM1 E3 ligase components, together with NEMF, under ANS conditions. We also plotted ribosomal proteins and do not see a clear enrichment of either 40S or 60S proteins (Response Figure 2, middle panels). Using ribosome pellets as input in the pull-downs results in high background binding of the concentrated ribosomes to the beads. To overcome this caveat, we also used lysates as input for FLAG-

UFM1 IPs (Figure 1D, E, Figure EV1A). We observed a clear enrichment of RPL26-UFM1 relative to the background, whereas small-subunit proteins were not detectable in the eluates (Figure 1E, Figure EV1A). Importantly, we observe a ribosome-stalling-dependent association of NEMF and Listerin in FLAG-UFM1 pulldowns under these conditions (Figure 1E, Figure EV1A), validating our findings. Please also see the lines 170-183.

Response Figure 2. Volcano plots showing enrichment of proteins in FLAG-IPs compared to the IgG controls in ANS conditions.

1c) A pulldown experiment, as in Fig. 1B, should be blotted for all the UFMylation and RQC factors (along with 60S and 40S markers) and Sec61 to validate the MS analysis.

We thank the referee for the input and the suggestion. During revision, we have performed FLAG pulldowns from FLAG-UFM1-expressing cell lines, showing specific enrichment of UFM1-RPL26 conjugates and the association of NEMF/LTN1/UFL1 in a stalling-dependent manner, please see Figure 1D, E, Figure EV1A. We also performed a pulldown of Sec61 translocon and observed specific enrichment of UFMylated RPL26 upon ribosomal stalling, indicating that UFMylation of RPL26 happens on translocon-docked peptidyl-tRNA-60S complexes (Figure EV1B). Please also see the lines 170-183.

1d) There are a number of aspects of Fig. 1C that are confusing to me. According to the data shown in panel B, the cushion-normalised ANS/Unt ratio should be 1 if the point falls on the diagonal, would be higher than 1 if it is above the diagonal, and lower than 1 if it is below the diagonal. Yet, DDRGK1, which is clearly below the diagonal in panel B, is shown in panel C to have a ANS/Unt ratio of 4 in both replicates. Similarly, CDK5RAP3 is only a little above the diagonal and nowhere close to NEMF or UFM1 in panel B, yet in panel C, is shown to be roughly the same in both replicates. Thus, the data shown in B and that depicted in C do not seem to match, even though the legend indicates that panel C is a bar graph using data from panel B plus another replicate. Looking at the supplementary table also seems to indicate that

what is plotted in panel C is not normalised to the cushion samples. Furthermore, it seems that the same untreated sample is used as the denominator for the two replicates, indicating that what is plotted in panel C is not actually two independent replicates. All of these things need to be clarified and the analysis should be more transparent and rigorous.

We now performed the experiments in triplicate to produce volcano plots, a gold standard in the field. We now show enrichment over control IgG and the corresponding p-values. In these experiments, we observe clear enrichment of all components of the UFM1 ligase machinery (UFL1, DDRGK1, and CDK5RAP3) and NEMF in the FLAG-UFM1 pulldowns (Figure 1B, C).

2) The data in Fig. 2 basically reproduces earlier experiments, and it is important that the authors acknowledge this. In particular, the fact that UFMylation is needed for effective degradation of an ER-RQC substrate but not a cytosolic RQC substrate was nicely documented in PMID 37036982 and 31595041. Furthermore, the heading "RQC factors and UFMylation machinery collaborate..." is not supported by what is shown in Fig. 2. The data only allow one to conclude that both sets of factors are needed for maximal degradation, but there is nothing to indicate that they collaborate. Please adjust the language here.

We thank the referee for their input. We cited these papers throughout the manuscript and checked whether we were missing them at crucial points, and added them. Moreover, we adjusted the language in the heading at line 225 accordingly.

3) There are a number of places where the authors blur the lines between what their data show versus what they infer from earlier work to interpret their data. For example, starting on line 270, they say they show that stalled ribosomes are recognised by ZNF598 and split by the ASC-1 complex, but in fact, they never show this. Instead, this is inferred from earlier studies that investigated this, and this information is then used to conclude that Ufmylation depends (partially) on splitting. What is actually shown by the authors is that Ufmylation is diminished in the absence of either ZNF598 or ASC-1 complex. The implication of this observation is that recognition of collisions (mediated by ZNF598 - PMID) and subsequent ribosome splitting (mediated by ASC-1 complex - PMID) are likely to be prerequisites for Ufmylation. I strongly encourage the authors to be far more precise in their language and interpretations.

We thank the referee for their feedback. We have now corrected the text and cited the referred papers to avoid confusion. Please see the lines 266-271 and 319-322.

4) In Fig. 3E, it is stated the ERK20 increases Ufmylation, whereas CytoK20 does not. But in this experiment, there is no sample for cells not expressing a stalling reporter. So it is unclear to me what is being taken as baseline here. Perhaps the authors mean to say that ERK20 shows higher Ufmylation compared to CytoK20? Please clarify.

In this experiment, as the referee mentioned, we compared the impact of CytoK20 expression to that of ERK20 on RPL26-UFMylation. We have now included cells that do not overexpress a

reporter as background controls (Response Figure 3), and we do not observe an effect of CytoK20 expression on RPL26-UFMylation, as shown previously ³.

Response Figure 3. ERK20 increases RPL26 UFMylation. UFMylation levels are visualized by UFM1 immunoblot of HCT116 cell lysates upon 24 h transfection with ERK20, CytoK20, or no transfection as control. FL – full length, AP – arrested peptide, g – glycosylation, # - UFC1-UFM1 complex, * - degradation products.

5) The authors often make definitive statements when they are not warranted. For example, they say splitting is required for Ufmylation, but there is no solid evidence to support this claim. Instead, they see a PARTIAL reduction in Ufmylation when the ASC-1 complex or ZNF598 are absent. They need to acknowledge this partial effect, and ideally, provide some explanation for why the effects are partial. Again, as in the previous comment, more care is needed in how the data are interpreted and how the paper is written.

Recent work showed that RPL26-UFMylation also occurs after canonical translation termination at the ER, releasing free 60S from the translocon ^{4,7}. We anticipate that the RPL26-UFMylation at steady state conditions is governed by two main events: translation termination at the ER following ribosome splitting via translation termination factors, and after ribosomal stalling at the ER following the splitting via ASC-1 complex or Pelota, depending on the type of mRNA. Therefore, depletion of ASC-1 or ZNF598 has only a partial effect. We have now added this interpretation to the text, please see the lines 266-271.

6) The effect of Baf is more substantial for the ER reporter than on the cytosolic one. Perhaps mention this observation, and the possibility that the ER reporter can be potentially released into the secretory pathway and trafficked to lysosomes for degradation. Also, Baf is not really an autophagy inhibitor; it is better to say it is an inhibitor of lysosomal degradation.

We thank the referee for bringing this to our attention. We have now corrected this mistake in the text (lines 245-249).

7) Pelota knockdown sometimes decreases Ufmylation, but other times has no effect. Can the authors please clarify the reason(s) for this.

We observe a clear effect of Pelota depletion on the accumulation of RPL26-UFM1 in plants, whereas in mammalian cells this effect is much less pronounced. We speculate that the composition and activity of various RNases in plants might generate truncated mRNAs and lead to ribosomal stalling at the 3'-end, which is primarily cleared by Pelota. We have now included this explanation in the text, please see the lines 314-317.

8) The substrate pulldown experiments in Fig. 4 should each be analysed more thoroughly by blotting the samples for other relevant factors in addition to what is shown: 40S protein, NEMF, and LTN1. Furthermore, there is no control sample lacking a GFP-tagged substrate, so one cannot judge what background recovery would look like. The experiment as presented is rather incomplete and does not allow for unambiguous conclusions.

We now probed for 40S proteins, NEMF, LTN1 in pulldown experiments. We use ERK0 as a control in those pulldowns compared to the ERK20 stalling reporter. This is the closest control one can achieve with a similar protein sequence and ER targeting. In our experience, using affinity capture beads with no specific binding increases background interactions due to increased surface area for non-specific interactions. We now show ERK0 control for all the interaction partners we probed in various figures.

9) The legends of Fig. S1 don't match what is shown in the figure. The legend has panels A-E, the figure only panels A-D; also, some of the panels don't match the legend. Please fix.

We thank the referee for pointing this out. We have now corrected this.

Referee #3:

This manuscript demonstrates the interplay between the ribosome-associated quality control (RQC) machinery and UFMylation in the clearance of arrested peptides at the endoplasmic reticulum (ER). The key findings are that dissociation of the 80S ribosome into the 60S subunit is a prerequisite for UFMylation of RPL26, a component of the 60S subunit, and that the UFM1 E3 ligase complex subsequently facilitates the recruitment of late RQC factors NEMF and LTN1. The experiments are well-designed, and the results are clearly presented. However, the majority of these core findings have already been reported (PMID: 38383785, PMID: 38383789, PMID: 40315331), and the novelty here is largely limited to implicating early RQC factors in the process. Moreover, the involvement of these early factors has already been anticipated based on prior studies. While I appreciate that this study strengthens prior structural observations with additional genetic and biochemical evidence, the overall impact of the findings is limited. In its current form, I think the manuscript does not meet the novelty and significance criteria for The

EMBO Journal.

We thank the referee for the kind words on our experimental design and presentation. We agree with the referee that genetic depletion experiments indicated that RQC is involved in the clearance of ER-targeted stalling reports. Yet whether RQC acts independently of the UFMylation machinery remained unclear, and how UFMylation clears stalled ribosomes has been debated. Here, we showed that close coordination between RQC and UFMylation enables the clearance of arrested polypeptides at the ER and revealed the sequence of events, thereby allowing us to dissect the roles of each pathway in this coordinated action.

Minor Comments

1. The authors should indicate RPL26 and LTN1 in Figure 1B.

We thank the referee for the input; unfortunately, LTN1 is very lowly expressed in HCT116 cells, so we do not detect it in the MS analyses. Similarly, although we observe clear enrichment of RPL26-UFM1 in the WB analyses, we do not detect it in the MS analyses. To address this point, we have now added FLAG-UFM1 pulldown WB analyses in Figure 1D and 1E (Figure EV1A) for two different cell lines, HCT116 and RKO.

2. The authors should cite the work from the Beckmann lab (PMID: 40315331), which demonstrated that the UFM1 E3 ligase complex forms a complex with the late RQC factors NEMF and LTN1.

Indeed, when our manuscript was under revision, a paper from the Beckmann/Kopito labs was published demonstrating that the UFM1 E3 ligase complex associates with the RQC factors NEMF/LTN1, validating and strengthening our findings. We now cite that in the text and in the discussion, please see the lines 501-507.

Methods related to the Response Figure 1

Establishment of LTN1 rescue cell lines

LTN1 rescue constructs were introduced into RKO iCas9 LTN1 KO (doxycycline-inducible) cell line using lentiviral transduction. Human LTN1 sequences 1-1766 aa for WT and 1-1712 aa for Δ RING mutant were cloned into the plenti-CAG-IRES-GFP plasmid (Addgene #69047) using XhoI and EcoRI restriction sites and Gibson assembly. RKO iCas9 LTN1 KO with integrated plenti-CAG-IRES-GFP construct was used as a control cell line.

RNA isolation and RT-qPCR

Total RNA was isolated using the KingFisher Flex Purification System (Thermo) with the High-Performance RNA Isolation kit (Molecular Tools Shop, Vienna BioCenter). During the isolation RNA was treated with DNase I (M0303S, NEB). cDNA was prepared with LunaScript RT

SuperMix (NEB) and amplified in a qPCR reaction with 2x Hot Start qPCR master mix (Molecular Tools Shop, Vienna BioCenter) using BioRad CFX 384 Touch machine. The levels of endogenous human *LTN1* transcript were assessed using the following primer pairs: 5'UTR (F:5'GAAGTTGTGTCCCGGACGTG'3, R: 5' TTTGAAGGCCTCAGGTTCCC '3) and 3'UTR (F:5'AGTCCAAGTGAAGCAAACGA'3, R: 5' TTCAGAGCATGCCTCCAGAT '3) and normalized to *RPL6* (F:5' ATTACGGAGCAGCGCAAGAT'3, R: 5' AACACAGATCGCAGGTAGCC '3).

References

1. Bengtson, M. H. & Joazeiro, C. A. P. Role of a ribosome-associated E3 ubiquitin ligase in protein quality control. *Nature* **467**, 470–473 (2010).
2. Udagawa, T. *et al.* Failure to Degrade CAT-Tailed Proteins Disrupts Neuronal Morphogenesis and Cell Survival. *Cell Reports* **34**, 108599 (2021).
3. Wang, L. *et al.* UFMylation of RPL26 links translocation-associated quality control to endoplasmic reticulum protein homeostasis. *Cell Research* **30**, 5–20 (2020).
4. DaRosa, P. A. *et al.* UFM1 E3 ligase promotes recycling of 60S ribosomal subunits from the ER. *Nature* 1–8 (2024) doi:10.1038/s41586-024-07073-0.
5. Simsek, D. *et al.* The Mammalian Ribo-interactome Reveals Ribosome Functional Diversity and Heterogeneity. *Cell* **169**, 1051-1065.e18 (2017).
6. Gak, I. A. *et al.* UFMylation regulates translational homeostasis and cell cycle progression. 2020.02.03.931196 Preprint at <https://doi.org/10.1101/2020.02.03.931196> (2020).
7. Makhlof, L. *et al.* The UFM1 E3 ligase recognizes and releases 60S ribosomes from ER translocons. *Nature* 1–8 (2024) doi:10.1038/s41586-024-07093-w.

Dear Dr. Karagöz,

Thank you for submitting a revised version of your manuscript. Your study has now been seen by all original referees, who find that their previous concerns have been addressed and now recommend publication of the manuscript. There remain only a few mainly editorial points that have to be addressed before I can extend formal acceptance of the manuscript:

- On the abstract page of the manuscript, please include 4-5 general keyword terms to enhance searchability.
- Please rename the "Conflict of Interest" section into "Disclosure and Competing Interests Statement", in accordance with our updated Guide to Authors (<https://link.springer.com/partners/embo-press/editorial-policies#Competing%20interest%20disclosures>)
- AUTHORS: author name discrepancy - G Elif Karagöz in the manuscript vs. Gülsün Elif Karagöz in the system (the author should provide his full first name in the manuscript file, if possible)
- Please adjust the format of the reference list and of the in-text citations according to EMBO Journal format (alphabetical order, author name et al + year.../up to 10 author names in the reference list before et al / please refer to our Guide to Authors for additional information on EMBO J reference format).
- FUNDING INFO: Chemical Biology is listed in the Acknowledgements. but it is missing in the system
- DATASET EV LEGENDS: Supplementary table 1 and Supplementary table 2 are datasets and need to be updated to Dataset EV1 and Dataset EV2 in all places (source file names, titles in the system, legends, callouts in the manuscript); the legends need to be removed from the manuscript and each should be provided in a separate tab/sheet in its corresponding Excel file
- Please provide suggestions for a short 'blurb' text prefacing and summing up the conceptual aspect of the study in two sentences (max. 250 characters), followed by 3-5 one-sentence 'bullet points' with brief factual statements of key results of the paper; they will form the basis of an editor-written 'Synopsis' accompanying the online version of the article. Please also provide an altered synopsis image, making sure that the aspect ratio conforms to our website's format - it should be exactly 550 pixels wide and between 300-600 pixels high.
- Please provide the Reagent and Tools Table. For more information, please check <https://media.springernature.com/original/springer-cms/rest/v1/content/27825802/data/v1>
- SOURCE DATA: SD provided with completed checklist; each folder should be uploaded separately
- The manuscript sections should be in the following order: Title page - Abstract & Keywords - Introduction - Results - Discussion - Methods - Data Availability - Acknowledgments - Disclosure Statement & Competing Interests - References - Figure Legends - (Main Tables with legends if applicable) - Expanded View Figure Legends.
- Please correct the Nomenclature for EV figures. It should be Figure EV1, etc. instead of Expanded View Figure 1, etc. (source file names, titles in the system, legends, callouts in the manuscript)
- BioRender should be acknowledged at the end of the Methods section in the following way:
Graphics:
(some of the... OR Figure #... OR synopsis) Graphics were created with BioRender.com.
- Figure Legends:
 - 1) Please note that the legends for figures 2, 3 is not provided in the sequential manner.
 - 2) Please note that the figure EV4 is mislabeled as figure EV5 in the manuscript.
 - 3) Please define the annotated p values ****/***/**/* as well as provide the exact p-values for the same in the legend of figure 2E, 3B, D, F, I, M; 4E, G, J; 5B, D, E; EV3 B as appropriate.
 - 4) Please indicate the statistical test used for data analysis in the legends of figures 1B, C; 2E, G; 3B, D, F, G, I, K, M; 4E, G, I, J; 5B, D, E; EV3 B, D
 - 5) Please note that information related to n is missing in the legends of figures 1B, C

6) Please note that the error bars are not defined in the legends of figures 2E, G; 3B, D, F, G, I, K, M; 4E, G, I, J; 5B, D, E; EV3 B, D

With best regards,

Cornelius Schneider

Cornelius Schneider, PhD
Editor | The EMBO Journal
c.schneider@embojournal.org

Please refer to our figure preparation guideline in order to ensure proper formatting and readability in print as well as on screen:

<https://link.springer.com/journal/44318/submission-guidelines#cms-Figure-and-data-presentation>

Referee #1:

The authors have satisfactorily addressed the reviewers' comments, strengthening their conclusions while uncovering unexpected new data. I support publication of the manuscript in its current form.

Referee #2:

The authors have done additional work to better support their conclusions, and have improved the writing to better place their findings in context. I support publication of this work.

Referee #3:

I think the authors have made a substantial effort to address the reviewers' comments, and the revised version is now technically solid and much improved.

My remaining concern mainly relates to the level of conceptual advance relative to recent publications in this area, rather than to scientific rigor per se. I defer to the editor's judgment as to whether the current level of conceptual advance meets the journal's criteria.

The authors have addressed all minor editorial requests.

Dear Dr. Karagöz,

I am pleased to inform you that your manuscript has been accepted for publication in the EMBO Journal.

You may qualify for financial assistance for your publication charges - either via a Springer Nature fully open access agreement or an EMBO initiative. Check your eligibility: <https://link.springer.com/journal/44318/how-to-publish-with-us>

Yours sincerely,

Cornelius Schneider, PhD
Editor
The EMBO Journal
c.schneider@embojournal.org

Please note that it is The EMBO Journal policy for the transcript of the editorial process (containing referee reports and your response letters) to be published as an online supplement to each paper. If you should prefer removal of any referee-only figures included in the point-by-point response(s), e.g. because they may still be used for future publication or because they have been reproduced from published work by others, please do let us know immediately via response email.

More information is available here: <https://link.springer.com/partners/embo-press/editorial-policies#Peer%20review>